# Early indicators of tidal ecosystem shifts in estuaries

Gregory S. Fivash [1,2] ✉, Stijn Temmerman [3], Maarten G. Kleinhans [4], Maike Heuner[5], Tjisse van der Heide [2,6,7] & Tjeerd J. Bouma[1,2,4,8]

Forecasting transitions between tidal ecosystem states, such as between bare tidal flats and vegetated marshes, is crucial because it may imply the irreversible loss of valuable ecosystem services. In this study, we combine geospatial analyses of three European estuaries with a simple numerical model to demonstrate that the development of micro-topographic patterning on tidal flats is an early indicator of marsh establishment. We first show that the development of micro-topographic patterns precedes vegetation establishment, and that patterns tend to form only on tidal flats with a slope of <0.3 degrees. Numerical modelling then provides an explanation for the formation of micro-topography due to the natural concentration of draining surface water over very gentle slopes. We find this early indicator to be robust across three estuaries where anthropogenic deepening and narrowing has occurred in recent decades, which may suggest its broader applicability to other estuaries with similar morphological management.

Tidal ecosystems are highly valued for their disproportionate role in the provisioning of ecosystem services, which range from flood risk mitigation and carbon sequestration to nursery functions for fish-stocks[1]. And yet, all over the world tidal ecosystems are impacted by both global change processes (such as sea level rise[2]) and regional anthropogenic activities (such as land reclamation[3]) that are causing shifts in their functioning. The trajectory of tidal flats is seen as a major societal concern, with repercussions ranging from flood safety to the loss of biodiversity[4–6]. The counter-balancing forces of tidal flat build-up and break-down, driven by currents and waves, can be strongly impacted by anthropogenic coastal development[7,8]. In many estuaries that service inland ports, the port authorities have overseen the progressive deepening of central-navigation channels to facilitate ever-larger vessels[9,10]. Estuaries have also commonly become narrower as a consequence of encroaching land reclamation of what were originally intertidal areas[11]. What results is a deeper and narrower estuary, which

magnifies the tidal amplitude and current strength[8,12,13], while also curtailing wind-driven wave generation within the estuary, due the shortening fetch length[7,14,15], offset to some degree by the increasingly role of erosive ship wakes[16]. These hydrodynamic changes tend to bias build-up processes on the tidal flats, gradually transforming steadily sloping gradients into more binary systems with a sharp transition between high intertidal flats and a deep central channel[8].

Geomorphological changes may incite further ecological transitions that affect the character of estuaries and the ecosystem services they provide. A transition from, for instance, a bare tidal flat to a vegetated marsh may be considered desirable from a coastal protection perspective[17,18]. But the resulting loss of tidal flat habitat comes at the cost of their own unique set of services, such as the provisioning of food-rich stopovers along migratory bird flyways[19]. Given this trade-off in services, coastal management would benefit greatly from the ability to foresee on-coming ecosystem transitions before they begin. In any

[1]Department of Estuarine and Delta Systems, Royal Netherlands Institute for Sea Research, Yerseke, The Netherlands. [2]Groningen Institute for Evolutionary Life Sciences, Community and Conservation Ecology Group, University of Groningen, Groningen, The Netherlands. [3]Ecosystem Management Research Group, University of Antwerp, Antwerp, Belgium. [4]Department of Physical Geography, Faculty of Geosciences, Utrecht University, Utrecht, The Netherlands. [5]Department of Vegetation Studies and Landscape Management, Federal Institute of Hydrology, Koblenz, Germany. [6]Aquatic Ecology and Environmental Biology, Institute for Water and Wetland Research, Radboud University, Nijmegen, The Netherlands. [7]Department of Coastal Systems, Royal Netherlands Institute for Sea Research, Den Burg, The Netherlands. [8]Delta Academy Applied Research Centre, HZ University of Applied Sciences, Vlissingen, The Netherlands. ✉e-mail: greg.fivash@nioz.nl

of these scenarios, the geomorphological development of tidal flats, through sedimentation and erosion processes, commonly occurs gradually over a decadal timescale while the ecological transitions motivated by those changes tend to take place within the span of only a few years. For this reason, ecological transitions are commonly described using the alternative stable state theory[20,21], with the transitions from tidal flats to vegetated marshes often referred to as a clear example[22–24]. The self-reinforcing nature of ecological transitions through feedback mechanisms makes them difficult to prevent once they have begun[20,25]. Because of this, the management of estuaries should make better use of indicators that provide an early indicator before an ecological transition is triggered, rather than indicators that signal the onset of a transition, itself. In this study, we search for such early indicators using publicly available datasets to better understand what drives the onset of vegetation establishment on initially bare tidal flats in three European estuaries heavily utilized as shipping-lanes.

Marsh vegetation is naturally constrained to the upper regions of the intertidal zone that do not flood during at least a fraction of high tide periods (above the neap high water level), due to fluctuations of high tide levels[26]. This natural limit is understood to be a consequence of unique seedling establishment dynamics on tidal flats, in which periods of refuge from inundation eliminate the risk of exposure to hydrodynamic disturbances by currents and waves[27]. These temporal refugia (known as windows-of-opportunity[28]) allow young pioneer plants, vulnerable to disturbance, to persist and develop into larger more resistant vegetation[29]. As tidal flats grow higher by sediment accretion, periods of refuge from inundation invariably increase in frequency, multiplying the opportunities for marsh pioneers to colonize the tidal flat[30]. While there exists a clear elevation threshold below which marsh vegetation cannot persist, the actual extent of marsh vegetation up to that threshold can vary dramatically due to other local variables[26].

Presently, studies aiming to predict the occurrence of intertidal vegetation rely mainly on the topographic elevation and local hydrodynamic forces[22,30,31]. However, small-scale morphological features on the tidal flat known as micro-topography have been found to create habitat uniquely suitable to the establishment of pioneer vegetation, in small-scale studies[32,33]. When micro-topographic features appear, the normally smooth surface of the tidal flat is restructured into a repeating pattern of meter-scale ridges and runnels, with a vertical relief usually no more than 10 cm[34,35] (Fig. 1). Within these patterns, the higher ridges experience subsurface drainage in the top centimeters of the sediment over the low-tide interval, while the lower areas often remain permanently submerged in pooling water. The enhanced subsurface drainage in the ridges of micro-topographic patterns results in more substantial sediment porewater recirculation, driven by tidal cycles of drying and re-wetting[36]. This re-circulation provides sediment oxygen, which reduces the concentration of sulphidic soil toxics in anoxic marine sediments[37,38] and thereby benefits pioneer growth[36]. Vegetation also likely benefits from the reduced erodibility of raised micro-topographic surfaces[29].

The establishment of vegetation in association with tidal flat micro-topography has been the focus of a growing number of studies[32,33,39,40]. But to date, most of these studies focus on either de-embanked areas (polders that have been converted back to wetlands by being reintroduced to tidal inundation), or experiment with artificially created micro-topography in order to develop applications for wetland restoration. Meanwhile, the question of where and when these features tend to form naturally remains mostly unexplored. This has had the unintended consequence that the widespread occurrence of these patterns in natural settings has been downplayed and their role in large-scale natural vegetative expansion events, overlooked. Yet micro-topographic patterns may be a setting uniquely capable of driving the transition from bare to vegetated tidal flats through the widespread facilitation of marsh seedlings, rather than the more common, but much slower clonal propagation (as discussed in Fivash et al.[32] & van de Vijsel et al.[33]).

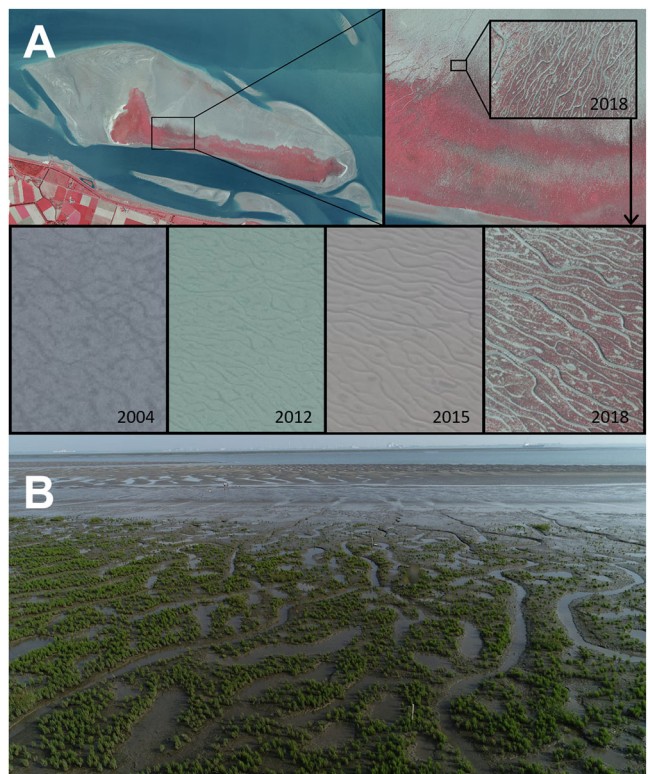

**Fig. 1 | A visual showcase of the role of micro-topographic patterns in vegetation establishment.** **A** A false color orthophoto mosaic featuring the mid-channel intertidal bar, Hoogeplaat, in the Western Scheldt (top left, coordinates: 51.392, 3.647). This intertidal area has transitioned from 0.2% vegetated (3 ha) in 2004 to 19.6% vegetated (290 ha) in 2020. Micro-topographic patterning developed on the tidal flat north of the marsh (top right), proceeding further marsh establishment in later years. The time series of orthophotos of the tidal flat (bottom panes) demonstrate the progression from unstructured tidal flat patterning (2004) to more intense channelized patterns (2012–2015), followed by marsh pioneer establishment (2018). In the bottom (**B**) the marsh pioneer Salicornia procumbens can be seen colonizing a nearby fringing tidal flat, Hooftplaat, where micro-topographic features are present (Photo credit: Jeroen van Dalen, NIOZ Yerseke).

To support this claim, we showcase here the role of micro-topographic patterns in vegetation transitions by first (1) identifying the conditions of pattern formation, and then (2) demonstrating how their formation is followed by the expansion of vegetation at the estuary scale. This is done through the analysis of remote-sensing data from the Dutch Western Scheldt estuary, available between 2004 and 2020, as well as two smaller datasets on the Humber and Elbe estuaries, in Britain and Germany. A simple numerical model is used to provide conceptual insight into the circumstances under which micro-topographic patterns form. This knowledge is then used to develop an early indicator of on-coming tidal flat-marsh transitions.

## Results

### Tidal flats flatten as they become higher across European estuaries

Geospatial analysis of lidar data from the Dutch Western Scheldt, German Elbe, and British Humber indicate that tidal flats above neap high water level in each of these estuaries have, on average, increased in elevation by 1–3 cm per year, over the last two decades (Fig. 2A). This has caused the tidal flats to experience more frequent periods of refuge from inundation (Fig. 2B). Tidal flats that reach higher levels in the tidal frame also appear to become flatter in a similar manner across all three estuaries (Fig. 2C).

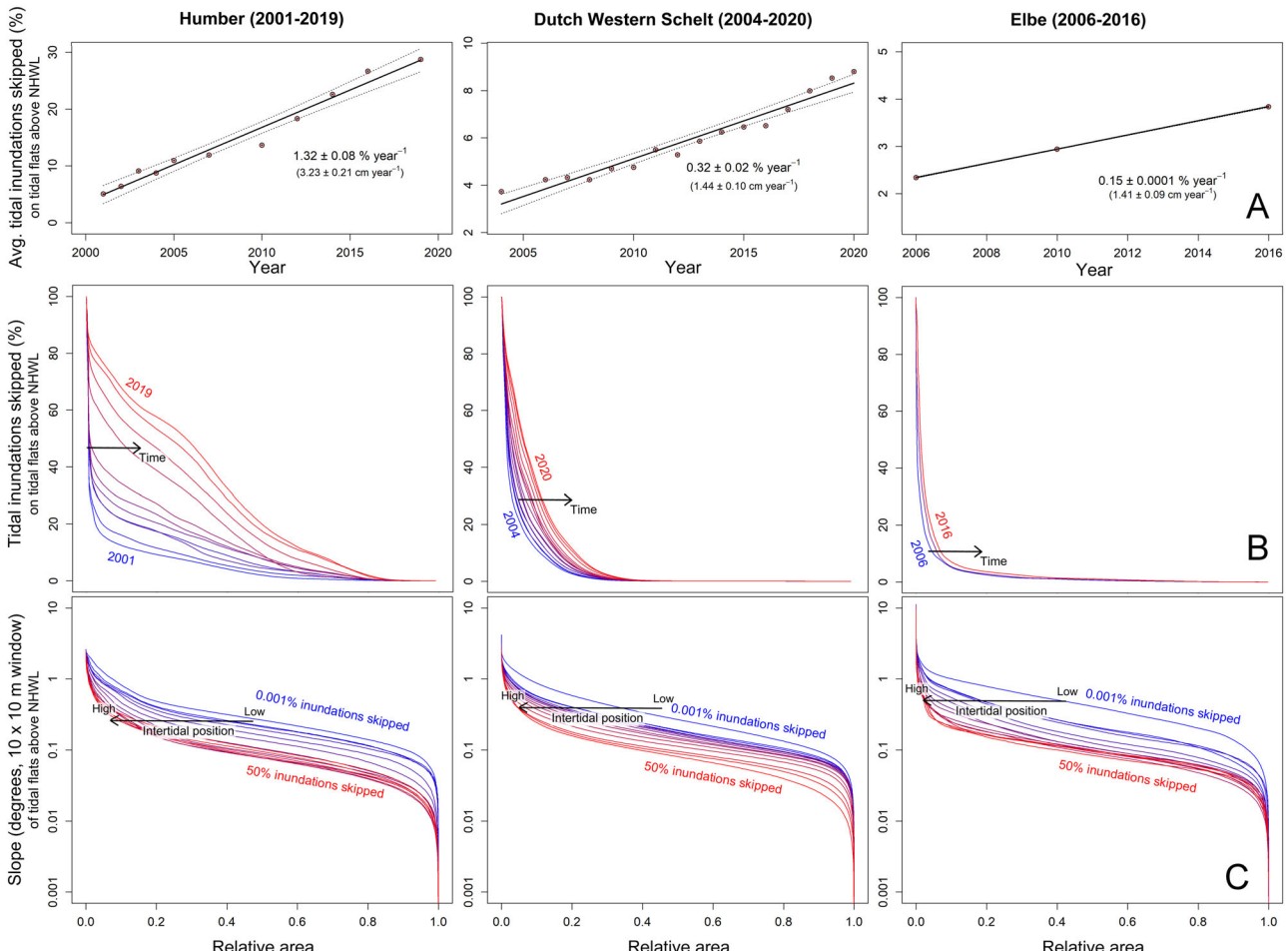

**Fig. 2 | Changes in upper intertidal flat morphology across three study estuaries. A** Linear regressions showing net increase in the average % skipped tidal inundations on intertidal flats above neap high water level (NHWL) in the study regions of the Humber (n = 11 years), Western Scheldt (n = 16 years), and Elbe (n = 3 years) for the years of available data. Also noted are the trends in cm yr$^{-1}$, calculated from independent fits based on raw elevation data. The fits of each regression are detailed in the supplementary appendix 3. Points represent the mean value, dashed lines represent 95% confidence interval, and error bars represent standard error around each point (the exact number of replicate measurements (n) recorded for each year can be found in the source data provided). **B** Hypsometric curves of % skipped tidal inundations for each study estuary, above NHWL. Each colored line represents a separate year of data (blue: oldest, red: most recent). Lastly, (**C**) hypsometric curves displaying the average slope of upper intertidal flats, binned into intertidal levels between 0.01% (blue) and 50% (red). These figures show that tidal flats in these estuaries tend to be flatter the higher they lie in the tidal frame. Together, this data suggests characteristic changes are taking place in the estuarine morphology, leading to a rising and flattening upper intertidal zone that will be preferential to marsh pioneer establishment.

## Pioneer vegetation establishment on micro-topographic patterns

Geospatial analysis of aerial false color orthophotos demonstrate that the total vegetated land area in the Western Scheldt increased by 607 ha between 2004 and 2020, at a rate of 38 ± 6 ha yr$^{-1}$, with similar patterns of net vegetation gain observed in the Humber and Elbe (Supplementary Fig. 1). This represents a massive shift from bare to vegetated area, with an increase from 22.5 to 29% of all tidal flats above neap high water level in the Western Scheldt becoming vegetated over the last 16 years. During this period, the likelihood of new vegetation establishment on previously uncolonized tidal flat has correlated with the strength of the micro-topographic structuring on the high intertidal tidal flats (Fig. 3). Although vegetation establishment still cannot occur beyond the tidal boundary at 0% tidal inundations skipped (Supplementary Fig. 2), more intense micro-topographic patterns facilitate establishment closer to that boundary, at lower position in the tidal frame (Fig. 3).

## Micro-topography forms on gently sloping upper intertidal flats

By using lidar data and false color images in combination to detect very small-scale topographic features, we determined that micro-topographic patterns are found predominantly on tidal flats with very

shallow slopes, peaking in intensity at around 0.1 degrees (Supplementary Fig. 3d). As with vegetation establishment, the intensity of tidal flat patterning is also consistently stronger in areas that experience more frequent relief from tidal inundation. (Supplementary Fig. 3c). However, no matter the elevation, micro-topographic patterns tend not to occur on tidal flats with a slope greater than 0.3 degrees (see Supplementary Fig. 3a). Thus, patterns can be expected to appear most often on accreting, convex upper intertidal flats, which feature flat anterior sections high in the tidal frame (see Supplementary Fig. 4 for an example).

## Micro-topography may arise from patterns in draining surface water

Simple simulations modelling the shallow flow of draining surface water over an inclined plane give insight into one explanation of why micro-topographic features occur in low-slope environments. In these simulations, the direction of flow over the plane is determined predominantly by the local direction of the steepest slope. Here, the local slope depends on the combination of two factors: (1) the underlying slope caused by the inclination of the plane, and (2) the local slopes caused by small irregular features imposed on the bed elevation. When the slope of the underlying plane is near zero, the small features in the

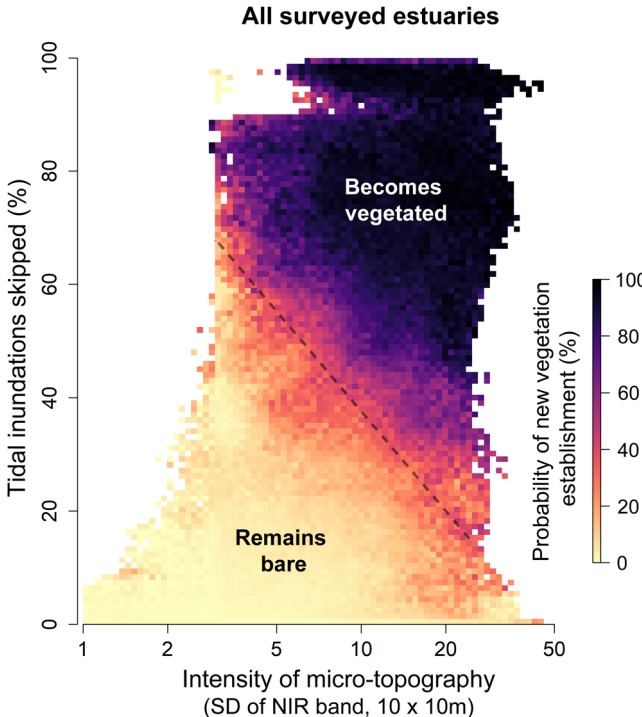

**Fig. 3 | Correlation between elevation, micro-topography, and vegetation establishment.** The probability of establishment of new vegetation between two consecutive measuring campaigns (1–3 yr intervals) is displayed against both (1) the frequency of skipped tidal inundations, and (2) the intensity of local micro-topography. All three sampled estuaries are combined, for all years of data available, to create this figure. Binned groups with $n < 30$ replicate measurements were excluded to eliminate noise in the image. Vegetation expands more frequently at higher position in the tidal frame, and generally cannot establish below neap high water, when 0% of tidal inundations are skipped. But areas with strong micro-topographic patterning support the establishment of vegetation at lower intertidal position, closer to that fundamental boundary (note diagonal edge of the dark vegetated area, marked by the dashed line).

bed elevation tend to cause water to pool at local elevation minima (Fig. 4). As the tilt of the underlying slope increases, water parcels begin to overcome small, raised features in the bed elevation and circumvent larger features in order to drain out of the inclined plane, leading to flow pathways that tend to link-up with others. This concentrates the flow of water over specific pathways of least resistance in the landscape. But if the underlying slope becomes too severe, the local slopes caused by small features in the bed elevation will eventually cease to be large enough to cause flow deviation in a direction other than that directed by the larger overall slope of the inclined plane. The result is that, depending on the magnitude of the irregular bed features, a certain tilt in the landscape will cause a shift from flow concentration in pools and drainage pathways to sheet flow where a similar amount of surface water flows down the plane everywhere. When we set the standard deviation of the irregular bed features to 2.5 cm (assuming a cell size of $1\,m^2$), the dynamics of our simulation appear to reflect the distribution of micro-topographic patterns in the geospatial data (Fig. 4A and B). Note that this model cannot explain the decrease in micro-topographic intensity at near-zero slopes observed in the geospatial data (see Fig. 4A).

## Discussion

Foreseeing transitions between ecosystem states is crucial, as they may imply irreversible loss of unique habits with highly valued ecosystem services. Transitions between tidal flats and marshes have been recognized as shifts between alternative stable ecosystem states

implying that they are difficult to reverse once they occur[22–24]. Despite this recognition, no practical early warning indicators have yet been able to forecast these transitions on large temporal (~decade) and spatial (~tens of km²) scales. However, the large geospatial datasets collected by government bodies and made available relatively recently, have made it possible to reveal that the micro-topographic development of tidal flats is a robust early warning indicator of this transition across several European estuaries. Whereas the transition from a bare to a vegetated state may be regarded as relatively rapid on a geomorphological timescale in line with the alternative stable state framework, at a smaller timescale we can detect a sequence of stages driving the transition. This begins with (i) the flatting of tidal flats, which drives (ii) the stabilization and intensification of micro-topographic patterning, which then (iii) facilitates vegetation establishment (Supplementary Fig. 4). It is this dissection into process-driven stages that allows the identification of early warning indicators that may prove crucial for estuary management. For this purpose, we propose here that accreting tidal flats above neap high water level can be considered to be approaching a transition to a vegetated state when they flatten in approach of the low-slope boundary, generally less than 0.3 degrees, at which micro-topography begins to appear.

The pattern of accretion on upper intertidal flats, which is ubiquitous across these three estuaries, is the fundamental driver of marsh expansion. It is most likely that this is a consequence of dredging practices, which enhance the tidal amplitude. When an estuary is dredged, the deeper channel bottom imposes less friction on tidal water entering and exiting the estuary. This enhances the flow velocity in the channel, which in turn leads to tidal amplification as the estuary width narrows along its course upstream[8,12,13]. Fast moving sediment-laden water quickly slows down when it moves onto the comparatively shallow tidal flat during high tide. As the water decelerates, a greater quantity of the sediment is deposited on the tidal flat nearest the channel[41]. This causes a greater increase in elevation near the channel than near land, which produces an increasingly convex tidal flat profile over time, with a flat anterior section[42] (see Supplementary Fig. 4 for an example). The result of the estuary-scale analysis in this study shows a clear correlation between the slope of the tidal flat and the intensity of micro-topographic patterning (Fig. 4A). The results of our inclined plane model suggest that the formation of micro-topographic patterns may be related to the formation of pools and the concentrating flow pathways of draining surface water in nearly flat environments (Fig. 4). While this model provides a robust explanation for why patterns do not form in steeper areas, it cannot provide a complete explanation of the conditions required for patterns to form. It cannot, for instance, explain the decrease in micro-topographic intensity in near-zero slope environments that is measured in the geospatial data. We have also found that micro-topographic patterning is more intense when the tidal flat is frequently above water during high tide (Fig. 4B). This suggests that the low tide period itself may play an important role in the amplification of these patterns, perhaps by exposing the sediment to prolonged periods of drying. It is also worth considering that the widening and heightening upper intertidal flat will also experience a reduced wave climate, which will reduce sediment turn-over[14]. Each of these various environmental characteristics in combination are likely important to the formation of permanent micro-topographic patterns. The major weakness in our understanding of these patterns up to this point is that has so far been guided exclusively by correlative evidence.

We still cannot yet provide a complete explanation of how and why these high, low-slope tidal flat environments tend to produce micro-topographic bed level variation, but we can provide some direction for further inquiry: The most obvious potential cause of these patterns is that the flow of draining surface water causes erosion in channeled areas. However, it is not obvious that the shear stresses caused by sheet flow over the extremely shallow gradients would be capable of erosion and then intensify these patterns (as suggested in

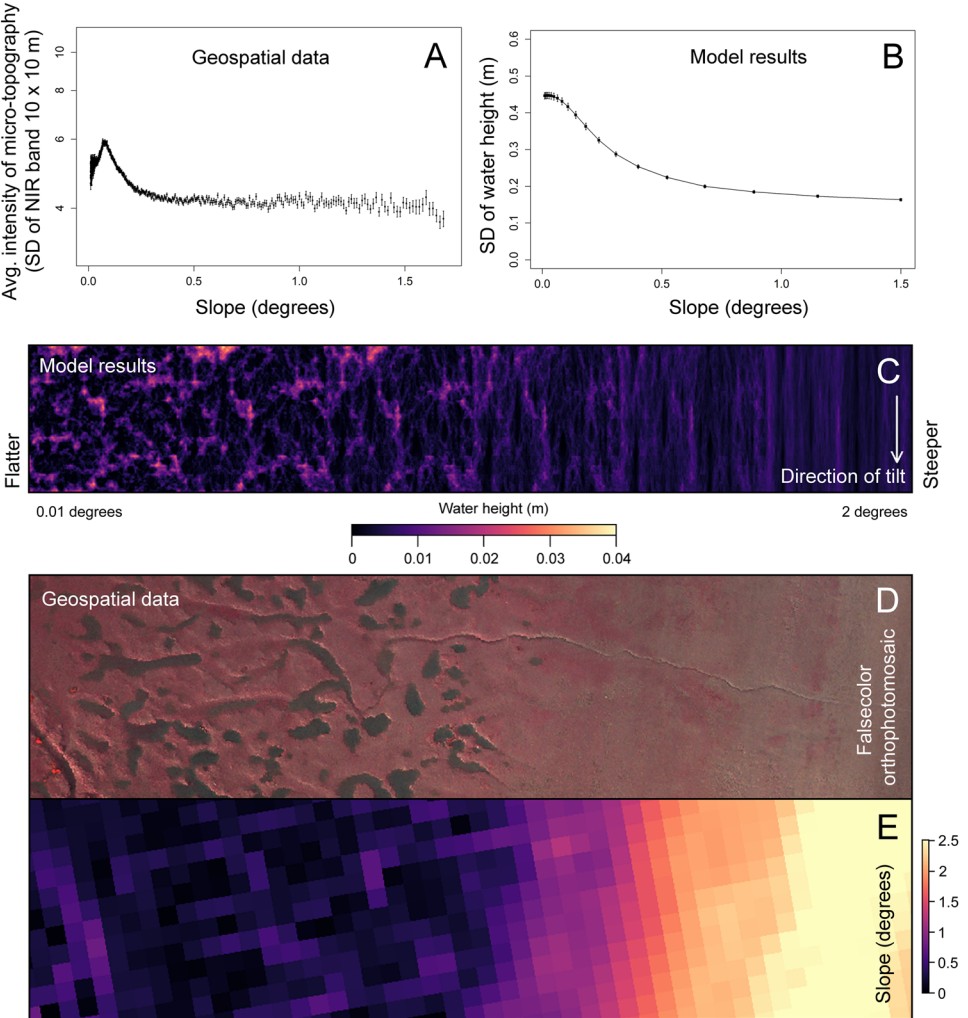

**Fig. 4 | Comparison of model results with the distribution of micro-topographic patterns in the geospatial data. A** Diplays the preferential occurrence of micro-topographic patterns on low-slope tidal flats considering all the three studied estuaries. Here, points represent the mean value of each binned group, and error bars show standard error (the exact number of replicate measurements within each binned group (n) for each mean can be found in the source data provided). **B** Displays flow concentration predictions produced by the inclined plane numerical model over planes of varying slope. Points show the average result across n = 100 simulations at each inclination. Each replicate simulation uses a unique randomly generated surface of autocorrelated topographic noise. Error bars represent standard error. **C** Visualizes how the distribution of surface water varies as a consequence of the underlying slope. Here the outcome of six simulations are displayed side-by-side, each with the same pattern of topographic noise, but with a steepening overall inclination from left to right. Note that in each of these panels, the plane inclines from top to bottom, so that the prodominant flow direction is from top to bottom. We compare the spectrum of simulated outcomes, ranging between a landscape of pooling water (**C**, left) and ubitqutous sheet flow (**C**, right) to geospatial data in the two lowest panels. Here, at the tidal flats near Hooftplaat, in the Western Scheldt, micro-topographic patterns containing pooling surface water abruptly end (**D**) when the slope of the tidal flat begins to steepen (**E**, 2 m res.).

Williams et al.[34] and Fagherazzi & Mariotti[35]). Another explanation is the concentration of surface water into small-scale local depressions affects sediment erodibility and hence results in increased bed level variability after erosive events. Likewise, the dewatering of raised ridges could accelerate the build-up of sediment cohesivity, and thereby reduce sediment erodibility on ridges[43]. There are also biological drivers that can amplify differences in erodibility. Microphytobenthos (chiefly diatoms) release EPS (extra-cellular polymeric substances) into the sediment in order to migrate vertically in the top millimeters of the sediment[44]. High sediment EPS concentrations enhance sediment cohesive strength[44], but the build-up of these substances appears to be hampered in water saturated sediments, particularly in pooling water[43,44,45]. Lower concentrations of EPS in the sediment in areas where draining water concentrates would also create differences in erodibility over space that would translate into greater bed-level variation through uneven patterns in erosion. Hence, once initially small elevation differences have developed, the patterns tend to intensify

through physical and biogeomorphic feedbacks[45,46] (Fig. 1) and ultimately prime the tidal flat for vegetation establishment[32,33,36].

What is to come of these changes? On its face, a net gain in vegetated foreshore area would appear to benefit flood safety[17,18,47]. However, in already narrow estuaries there is some risk that if marshes advance seaward, they would further raise the height of colonized flats by further enhancing sediment accretion[31]. The creation of high intertidal areas that are less often submerged effectively decreases the width of the estuary during a large period of the tidal cycle. This could cause the propagation and magnification of the tidal wave upstream[2,13], putting upstream communities at greater flood risk. This is somewhat ironic, given that wider marshes enhance flood security locally by reducing wave impact and run-up on the dikes behind them[47]. Greater marsh area will also come at the cost of feeding habitat for migratory birds in the estuary due to narrowing of the intermediate intertidal zones between the central channel and the marsh[19,48,49]. Altogether, an intertidal system composed pre-dominantly of high intertidal flats

bisected by a single deep central channel would host a poorer diversity of intertidal habitats and biodiversity[9,50]. Lastly, there may also be long-term risks associated with the development of the estuary in this direction. Studies in micro- and meso-tidal habitats have shown that large, vegetated marsh interiors are more prone to drowning[51–53], which may become a risk in macro-tidal estuaries once sea-level rise begins to accelerate in this region[2].

Finding solutions that will mitigate the impact of human activity on the estuary remains extremely difficult. There are two major current propositions to reduce the impact of dredging on estuarine processes: (1) moving inland ports to the coast so that dredging navigation channels is no longer necessary and (2) de-embanking sections of estuaries in managed-realignment projects to increase the effective width of the estuary and reduce the tidal amplitude[13,54]. Both come with obvious economic, political, and social costs that can be extreme. Nor does either proposition necessarily ensure a return to natural estuarine dynamics, as the seaward harbor could act as an obstruction to tidal flow[7], and managed re-alignment projects do not always create estuarine habitats comparable to natural analogs[55]. Suffice to say, finding an acceptable balance to the many trade-offs in the utilization of estuaries remains a worldwide challenge that will require much further expertise, deliberation, and funding to solve.

The development of biogeomorphic landscapes from bare to vegetated states has been discussed at length under two alternative frameworks: (1) community succession[56,57], and (2) alternative stable state theory[21,22,58–60]. In this example we see that a fusion of these two concepts best explains the transition. Here, a steady environmental change (in this case constant sediment accretion), eventually reaches a threshold at which a cascade of additional changes begins to take place. These secondary changes could be driven by physical mechanisms (like prolonged sediment de-watering) or by pioneering (micro-) organisms that appear prior to the vegetation transition. A similar case could be made for arid ecosystems where the development of a biologically active soil crust, which depends on adequate rainfall and limited erosion, is a critical precursor to the invasion of higher plants[61]. Apart from these, it is likely that similar stage-driven transitions occur in other ecosystems that have been described to follow alternative stable state dynamics. However, the lack of economic importance for most of these natural systems typically means that we lack the same level of high-quality large-scale temporal and spatial data that made this study possible. In estuaries that host major harbors, the costs of dredging to allow for navigation offer a major incentive for data collection. However, as technology develops and the costs of data acquisition decline (i.e., consider the increasing availability of UAVs[62]), producing similarly high-quality data should become a priority for developing a better understanding of less economically critical natural systems that nevertheless provide important and usually under-valued ecosystem services.

## Methods

### Data preparation: processing of public geospatial data
To perform geospatial analysis of estuarine development across three different European estuaries: The Dutch Western Scheldt, British Humber, and German Elbe, we analyzed aerial lidar elevation data, false color orthophotos, and water level time series collected and provided by Dutch, British, and German governmental agencies. Details on data acquisition and pre-processing specific to each estuary dataset can be found below in the final sub-section of the methods, (4) data acquisition and pre-processing.

**Calculating the frequency of skipped tidal inundations.** We used publicly available water level time series data to relate intertidal bathymetry to the frequency of tidal immersion in each estuary. When available, measurements from multiple stations were combined to properly capture changes in the tidal amplitude along the estuarys

length (details in the Supplementary Methods). These water level data were used to calculate the frequency of tidal inundations at every intertidal elevation by calculating the number of high tides that reached above a given elevation ($cycles_{inundated}$), divided by the total number of tidal cycles within the measured period ($cycles_{total}$). This produced a metric similar to the frequency of tidal inundations seen in Balke et al.[26]. However, in contrast to the inundation frequency metric used there, we then inverted the quantity to measure the percent of *skipped* tidal inundations (Eq. 2), which correlates positively with the probability of vegetation establishment (see Supplementary Fig. 2 for visualization).

$$\text{Tidal inundations skipped}(\%) = \left(1 - \frac{cycles_{inundated}}{cycles_{total}}\right) * 100\% \quad (1)$$

**Vegetation presence from false color orthophotos.** In order to track changes in vegetation cover, the red and near-infrared (NIR) bands of false color orthophotos were used to calculate the normalized difference vegetation index (NDVI)[63]. Vegetation was classified as present in pixels that exceeded an NDVI-threshold value. Because both different atmospheric effects and the timing of the photograph within the seasonal cycle varied across years, it was necessary to use a unique threshold value for each year, in each estuary, to accurately classify vegetation. Despite the fluctuations in absolute values, the overall distribution of NDVI values is consistently multi-modal in estuaries, clustering around specific values for water, sediment, and vegetation[31]. Thus, each NDVI-threshold was set by determining the average NDVI value of sediment and vegetation over the entire estuary in a given year using peak analysis, and then placing the threshold half-way between the two peaks (See Supplementary Fig. 5 for visualization). Details on the thresholds used per estuary and per year can be found in Supplementary Table 1. In analysis of the Elbe estuary, vegetation maps were already available via the German Federal Waterways and Shipping administration (WSV) and were used in place of the above NDVI characterization technique (see Supplementary Methods for details). Note that these three temperate European estuaries harbor little to no seagrass or sessile macroalgae, which otherwise would have complicated the classification of terrestrial vegetation. Benthic microalgae (e.g., diatoms), which are present, produce a positive signal in NDVI as well, however it is usually significantly lower than that of vegetation and differentiation is still possible with NDVI alone. An exception to this is fibrous epibenthic microalgae (e.g., *Vaucheria* sp.), which can occasionally be found in each of these estuaries, and it is not easily distinguishable from vegetation with NDVI alone. A certain amount of mischaracterization caused by problems like this one is undoubtably present in our study. However, the occurrence is rare enough to be very unlikely to have affected our main conclusions and the use of NDVI to characterize vegetated areas is standard practice for studies in this region[23,31,64,65].

**Detecting micro-topography using orthophotos.** A number of factors made it difficult to detect the presence of micro-topographic patterns in the tidal flat from topographic data alone. In some cases, the magnitude of height variation within a micro-topographic pattern was smaller than the vertical resolution of the lidar, rendering them invisible on bathymetry maps while the patterns were clearly visible in orthophotos (see Supplementary Fig. 6 for visualization). In other cases, the horizontal resolution of a pixel (often between 2 and 5 m) was too large to capture the structure of the micro-topographic patterns, which can have a pattern wavelength <3 m. Thus, in order to overcome these practical challenges, we chose to measure the presence and intensity of micro-topography in an unconventional manner, using orthophotos. The available orthophotos had a suitable horizontal resolution (between 12 and 25 cm) to clearly distinguish micro-topographic patterns. To measure the intensity of micro-topography on the tidal flat we calculated the standard deviation in

the NIR band within a $10 \times 10$ m frame. Repeating the same analyses using other spectral bands, or the greyscale image produced similar results. Thus, the fine cm-scale NIR orthophotos were transformed into coarser 10 m resolution images quantifying the variation in the reflectance of NIR light within a small region (See Supplementary Fig. 6 for visualization). All analyses in this study were performed on rasters resampled to a standard 10 m horizontal resolution in order to be comparable with our metric of micro-topographic patterns.

**Calibration of the micro-topographic intensity metric.** In order to test the reliability of using images to characterize geomorphological features on the tidal flat, we performed a calibration of our intensity of micro-topography (SD of NIR band) metric by comparing it to high resolution terrestrial lidar data. Between 2020 and 2022, we performed repeated terrestrial lidar surveys (RIEGL VZ-400i, RIEGL Laser MeasurementSystems GmbH, Horn, Austria) of three tidal flats in the Western Scheldt harboring micro-topographic patterns (Hoofdplaat: 51.373, 3.675; Zuidgors: 51.391, 3.855; and Baarland: 51.393, 3.866). The elevation point clouds produced by these surveys were converted to 25 cm resolution rasters matching the dimensions of the publically available orthophotomosaics. We then calculated the local slope of the tidal flat using the terrain function available in the R raster library[66]. We chose to measure variation in slope rather than the variation in elevation so that gradually sloping surfaces were not falsely labelled as having micro-topography. The orthophoto mosaics taken of the Western Scheldt in 2020, and 2022 were paired with lidar surveys that occurred nearest in time, all at least within 6 months of each other. The standard deviation of the (1) local slope and of the (2) NIR band intensity were measured in $10 \times 10$ m grids over the regional extent of the terrestrial survey. Then, these two metrics were compared using a linear regression after log-transforming both variables to normalize the distribution of the data. A strong correlation was found between these two variables, indicating that our approach for detecting micro-topography from NIR orthophotos is reliable when elevation data of suitable resolution is not available (Supplementary Fig. 7, $F_{1,1176} = 2966$, $R^2 = 0.72$, $p < 0.001$).

**Distinguishing micro-topographic patterns from other features.** Micro-topographic patterns are not the only features that are distinguished by the standard deviation of the NIR band. Vegetation and other bathymetric patterns, such as mega-ripples and tidal creeks, also return a strong signal with this metric. To focus the analysis solely on micro-topographic patterns, these other areas had to be excluded. Vegetated areas were masked using the NDVI approach detailed above. Micro-topography appears to form only on tidal flats above neap high water level, while mega-ripples generally occur in the lower intertidal and subtidal zones (See Supplementary Fig. 8 for visualization). Therefore, mega-ripples were excluded by including only areas where the percentage of skipped tidal inundations is greater than zero (described further simply as areas above neap high water level). Tidal creeks were excluded by masking areas with a topographic slope >3 degrees. The slope of the tidal flat was again calculated from the topographic data using the terrain function available in the R raster library[66]. Note that the slope of the tidal flat was calculated from the topography after being resampled to 10 m resolution, while creeks were identified first at 2 m resolution and *then* resampled to 10 m, thus the areas adjacent to creeks were also excluded from our analysis.

### Geospatial analyses
**Changes in intertidal elevation, slope, and vegetation cover.** The annual rate of change in the elevation of tidal flats above neap high water level was measured for each estuary with linear regressions (Fig. 2A). This was calculated separately for both (1) the average elevation and (2) the average percentage of skipped tidal inundations. Areas that were already vegetated in the earliest available year of data

(for each estuary) were removed from this analysis. Pixels with an NA value in any year (representing an area where elevation data was not collected) were excluded from the analysis in order not to introduce a bias into the data due to differences in the area measured between annual flights. The rate of change in the total vegetated area within the sampled regions of each estuary was also estimated using linear regressions (Supplementary Fig. 1). The fits of each of these regressions are detailed in Supplementary Table 2.

Changes in tidal inundation through time were also inspected through the creation of hypsometric curves[2,7,67]. Each hypsometric curve was created by plotting the percent of inundations skipped against the percentage of the intertidal area that equaled or exceeded each value (i.e., relative area), each year (Fig. 2B). Similarly, hypsometric curves were also used to inspect how the slope of the tidal flat changed depending on the position in the tidal frame. In this case, areas (aggregated from all years) were grouped into bins according to their tidal position (% skipped tidal inundations) and a hypsometric curve of the tidal flat slope was created for each bin (Fig. 2C).

**Effect of elevation & micro-topography on vegetation establishment.** Next, we performed an analysis to demonstrate the facilitative role of micro-topography in vegetation establishment that differentiated it from the impact of tidal inundation (similar to an analysis performed in Fivash et al.[33]). In this approach, we identified newly established vegetation by comparing adjacent years of vegetation data, generally spaced 1–3 years apart. Vegetation present in the later year of data, but not in the earlier, was considered to be newly established. The entire estuary was quantified (as described earlier) in terms of the two explanatory variables, (1) the intensity of micro-topography and (2) the percentage of skipped tidal inundations, which were then binned into groups according to these variables to create an $85 \times 101$ 2-D array with one variable on either axis (see Supplementary Fig. 9 for visualization). The likelihood of new vegetation establishment to occur within each of these binned groups was then determined as the proportion of cells in each bin that had become newly vegetated over that interval. This analysis was repeated for every pair of years, in each estuary, which were then averaged together to create a single 2D-array. Using the resulting 2D-array, we can distinguish the separate contributions of micro-topography and tidal inundation to new vegetation establishment. In addition to this, the average change in the vegetated area was determined for each of the three estuaries independently, using linear regressions (Supplementary Fig. 1). The fits of each regression are detailed in Supplementary Table 2.

**Role of elevation & slope in formation of micro-topography.** The relationship between the intensity of micro-topography (as measured by the SD of the NIR band) and the slope of the tidal flat across all three estuaries was considered by dividing the slope values logarithmically between 120 unique bins. Then, the average intensity of micro-topography was calculated within each binned group (Fig. 4a and Supplementary Fig. 3d). Note that as mentioned above, vegetated areas, tidal creeks, and areas below neap high water level were excluded prior to the analysis. The relationship between the intensity of micro-topography and frequency of skipped tidal inundations was also examined in the same manner, but in this case a linear regression was fit after log-transforming both variables (Supplementary Fig. 3c). All geospatial analyses appearing in this study were performed in R[68].

### Inclined plane numerical model
Here, we describe a simple numerical model that provides one possible explanation for the association between low-slope tidal flats and the appearance of micro-topographic drainage patterns. As our focus is on understanding the drainage patterns, we do not simulate full tidal cycles. Instead, we simulate the drainage of a thin layer of water, which can be considered as the drainage during ebb tide at the moment the

tidal flat starts to become dry after a tidal inundation. In these simulations, we follow the flow of water over an inclined plane to see how patterns in flow concentration change depending on the slope of the plane. A small amount of surface variability (SD: 2.5 cm, assuming a cell size of $1\,m^2$) is added to the plane in the form of auto-correlated noise (created through the fractal-folding method described by Saupe[69]). The model's purpose is to demonstrate how the concentration of draining surface water is affected by the slope of the underlying surface when minor local slopes caused by randomly distributed small topographic irregularities are also present.

The flow of water over the inclined plane is simulated using the flow equation described in Weerman et al.[45], expanded into two dimensions (Eq. 2). With this simple formulation, the velocity and direction of flow are determined by a combination of the water level and local slope of the plane, while maintaining conservation of mass and momentum.

$$\frac{dW}{dt} = \frac{d}{dxy}\left[KW\frac{d(W+S)}{dxy}\right] \qquad (2)$$

Here, $W$ and $S$ represent the local height (cm) of the overlying water layer ($W$) and the surface of the plane ($S$), respectively. Hydraulic conductivity ($K$), originally a function of water depth, is here set to a constant value equal to Weermans[45] original minimum value ($K = 1$) due to the very shallow water levels explored in this model. The simulation begins with a small initial quantity of water spread evenly over the plane ($W_i$). To ensure that the model consistently simulated the flow dynamics of very shallow surface water that would be re-directed by the topography of the tidal flat, the depth of this initial water layer ($W_i$) was set to equal 10% of the standard deviation of the autocorrelated noise. The water then flows over the plane until the average changing water level across the plane is very small (≤0.3% of $W_i$). In a series of repeated simulations ($n = 100$), the inclination of the plane was gradually increased over a range between 0 and 1.5 degrees. For each inclination in the series, the water level in each cell was recorded at each time step. In order to measure the extent to which the water tended to concentrate into specific regions or flow pathways, we calculated the average water height in each cell over the simulation and then measured the standard deviation of the average water height across the grid. This value, which we called the intensity of flow concentration (see Fig. 4B) was then compared between model runs with different underlying slopes.

These simulations used a $512 \times 512$ cell grid with repeating boundary conditions. Water parcels moving over the top-bottom boundary (returning to the top of the plane after flowing down it) continued to flow as if they had experienced a continuous gradient, via the following formulation:

$$S_{nx+1} = S_1 - \left(S_{max} + \frac{S_{max}}{nx}\right) \qquad (3)$$

$$S_0 = S_{nx} + \left(S_{max} + \frac{S_{max}}{nx}\right) \qquad (4)$$

Here, $S_{max}$ indicates the maximum height of the inclined plane and $nx$ is the length of the plane. $S_O$ and $S_{nx+1}$ represent neighboring cells that are beyond the extent of the grid ($S_O$ is before the first index, $S_{nx+1}$ is after the last index). These simulations were performed in Python version 3.8.10.

## Data acquisition and pre-processing
### Dutch Western Scheldt
**Geospatial data.** To perform geospatial analysis of estuarine development in the Dutch Western Scheldt, we collected publicly available elevation data derived from aerial lidar and false color orthophoto images provided by the Dutch Ministry of Infrastructure and Water

Management (Rijkswaterstaat), available at https://www.rijkswaterstaat.nl/apps/geoservices/geodata/dmc/. The elevation data is available in annual time series for the period from 2004 to 2020, at 2–5 m horizontal resolution and 1 cm vertical resolution. False color images (including NIR, red, and green bands, 12.5–25 cm resolution) were available in semi-annual time series for the years 2004, 2008, 2010, 2011, 2012, 2015, 2016, 2018, and 2020.

**Water level time series.** We used publicly available water level time series data from the same period (2004–2020, collected by Rijkswaterstaat, available at https://waterinfo.rws.nl/) to convert bathymetric elevation (measured in cm NAP) to the frequency of skipped tidal inundations. This was calculated by counting the number of high tides that reached above a given elevation and dividing that number by the total number of tidal cycles within the measurement period. Data from five stations (Westkapelle, Vlissingen, Terneuzen, Hansweert, and Bath) were combined to properly capture changes in the tidal amplitude along the estuary's length. First, the frequency of skipped tides was calculated for each unique elevation value in the intertidal zone, at each station. Then, for the areas in between the stations, the calculated frequency of skipped tides at each unique elevation were interpolated according to the longitudinal position in the estuary (the Dutch Western Scheldt runs predominantly east to west).

**Sampled area.** The intertidal bathymetric maps made available by Rijkswaterstaat are already masked at the subtidal and terrestrial boundary of the intertidal zone. These maps end at the Dutch-Belgian border, and thus miss the Belgian sector of the estuary between the border and Antwerp. The same regions masked in these bathymetric maps were then used to exclude terrestrial and subtidal areas from orthophotos.

### Elbe
**Geospatial data.** The same routine described above was used to process data on the Elbe estuary. The geospatial data was provided by the German Federal Waterways and Shipping administration (WSV, available at https://www.kuestendaten.de/DE/Services/Kartenthemen/Kartenthemen_node.htm).

In this case aerial lidar data (1 m horizontal, 1 cm vertical resolution) and false color orthophotos (25 cm resolution) were only available for the years 2006, 2010 and 2016. Biotope/vegetation polygon maps used to classify the vegetated area of the estuary were however also available for 2002, as well as 2006, 2010, and 2016. These polygon datasets were converted to a $10 \times 10$ m raster that indicated the presence or absence of vegetation, equivalent to those created using NDVI characterization in Humber and Western Scheldt. This was performed using the extract function in the raster library[66] to identify cells within an empty raster that were within polygons designated as representing either vegetated or unvegetated areas, based on their biotope classification. As such, orthophotos of the Elbe estuary were used exclusively to measure the intensity of micro-topography.

**Water level time series.** Water level time series were provided for 10 unique stations along the estuary (Otterndorf, Osteriff, Brunsbüttel, Brokdorf, Glückstadt, Krautsand, Kollmar, Grauerort, Stadersand, Hetlingen, originally collected by WSV (Federal Waterways and Shipping Administration, Wasserstraßen- und Schifffahrtsverwaltung des Bundes; available at https://www.kuestendaten.de/DE/Services/Messreihen_Dateien_Download/Download_Zeitreihen_node.html). These data were similarly interpolated along the longitudinal gradient.

**Sampled area.** Our study area began at the mouth of the Estuary near Neufelderkoog and ended at the provincial border between Schleswig-Holstein and Hamburg. Areas landward of the dikes and below the lowest measured low tide elevation (subtidal areas) were masked.

Note that in Germany some agricultural land sits seaward of the dikes and was thus included in our dataset. However, due to high variability in vegetation cover in these agricultural areas, when estimating the rate of vegetation change in the estuary, which is meant to highlight expansion of the pioneer zone toward the channel, we masked most agricultural areas by removing areas that skipped more than 90% of tidal inundations (see Supplementary Fig. 1).

**Humber**

**Geospatial data.** Geospatial data on the Humber estuary was provided by the British Department of Environment Food & Rural Affairs (via the Defra Survey Data Download portal, https://environment.data.gov.uk/DefraDataDownload/?Mode=survey). These data included aerial lidar data (1 m horizontal resolution and 1 cm vertical resolution, for the years 2001, 2002, 2003, 2004, 2005, 2007, 2010, 2012, 2014, 2016, and 2019) and false color orthophotos (20 cm resolution, for the years 2011, 2012, 2013, 2014, 2015, 2016, and 2018). To account for the misalignment of the data in select years, the bathymetry maps from the years 2010 and 2019 were considered to be contemporaneous with orthophotos taken in the years 2011 and 2018 during analysis requiring both datasets.

**Water level time series.** Water level time series from 1953 to 2011 were provided by the National Tidal and Sea Level Facilitys UK National Tide Gauge Network (https://ntslf.org/data/uk-network-real-time). Unfortunately, water level time series were only available at a single sampling station within the Humber (Immingham) so it was not possible to account for the amplification of the tidal range along the length of the estuary. Therefore, we constrained our analysis of the Humber to the tidal flats near the sampling station at Immingham so that calculations of skipped tidal inundations would be accurate.

**Sampled area.** The sampled intertidal area in the Humber ranged from northwest to southeast from the coordinates (53.736, −0.259) to (53.631, −0.0788). The landward extent of the intertidal zone was defined by the dikes, and the subtidal boundary was defined by the extent of the aerial lidar maps, which were masked in areas covered by water prior to distribution.

**Reporting summary**
Further information on research design is available in the Nature Portfolio Reporting Summary linked to this article.

## Data availability
The geospatial data and scripts used to perform the analyses that support the findings of this study are archived and publicly available via 4TU.Research Data, https://doi.org/10.4121/21762680. Source data are provided for Figs. 2–4, and supplementary Figs. 1–5, 7, and 9 as a Source Data file. Source data are provided with this paper.

## Code availability
The code used to run the inclined plane numerical model that supports the findings of this study is archived and publicly available via 4TU.Research Data, https://doi.org/10.4121/21762680.

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

## Acknowledgements

This research was financed through the German Federal Institute of Hydrology (BfG) in line with its R&D-Project Uferfunk (estuarine shoreline functions, M39630304077, G.S.F.). S.T. acknowledges financial support from the Research Foundation Flanders (FWO grant nrs. G039022N and G031620N). We thank the Dutch Ministry of Infrastructure and Water Management (Rijkswaterstaat) and the Province of Zeeland for funding studies on understanding the establishment of pioneer vegetation in the Perkpolder, Rammegors and Zuidgors projects. We also thank the department of Estuarine & Delta Sysytems for providing funding for the publication of this study.

## Author contributions

G.S.F and T.J.B. conceived the ideas. G.S.F designed the methodology, performed the analyses of results, and wrote and ran the simulations. G.S.F. and M. H. performed data acquisition. G.S.F., S.T., M.K., and T.J.B. led the writing of the paper. All authors contributed critically to the drafts and gave final approval for publication.

## Competing interests

The authors declare no competing interests.
