## [Peer Review File · Nature Communications]

REVIEWER COMMENTS

Reviewer #1 (Remarks to the Author):

This manuscript describes a study that primarily utilizes remote sensing data to determine how the ecological transition of intertidal mudflats to vegetated marshland can be detected through early warning first by the development of micro-topographic patterning. The researchers suggest that this early indicator of a transition occurs only on very shallow sloped tidal flats with a slope <0.5 degrees and occurs due to the impacts of a draining surface at elevations just above the neap high tide water level. A basic numerical model was used to show that at flatter surfaces with slopes near zero that pooling of water droplets occur at local elevation minima, while at larger slopes (>0.5 degrees), they flow down the incline along a direct path and tend not to generate flowpaths that link up to circumvent minor variations in the surface elevation.

Overall, I found this to be a very fascinating study that clearly has large implications for the fate of coastal mudflats and marshes in the face of sea level rise and other anthropogenic threats (including deepening of shipping lanes through dredging). The authors did a great job within the introduction of describing the relevant theory and benefits of determining early warning signs of ecosystem shifts and how it applies to intertidal mudflat/marsh systems. Their conclusions seem robust across 3 large estuary systems in Europe, and I believe this could be a seminal paper that will spur new research in to topographic influence on state-change dynamics. With the ever-increasing accuracy of digital elevations from lidar, GPS systems, and drone imagery, this will likely be an increasing area of research. The manuscript is very well written with a clear logical progression, and good use of figures and supplementary material. I do have one major concern and a few more minor ones that I feel need to be addressed.

The one major criticism I have is the application of how micro-topography was quantified and used broadly within this study. The authors chose to measure the presence and intensity of micro-topography through orthophotos that had a horizontal resolution of 25-50 cm. The magnitude or 'intensity' of micro-topography on a tidal flat was then calculated using the standard deviation in the NIR band within a 10m x 10m frame. No micro-topographic elevations were in fact measured at all. This may be a reasonable albeit 'novel' way to estimate micro-topography, but there was no mention within the methods of validating this method in any way. The entirety of their results rely upon the assumption that the standard deviation of the orthophotos are a good predictor of topography. Although I think this is a useful way to quantify micro-topography features using coarse data, it should be validated. Was any high resolution topography quantified over these small scales to compare to the standard deviations in the NIR band? This perhaps can be done using structure-from-motion analysis of drone flights to obtain higher resolutions, or more detailed gps surveys over smaller sections of a tidal flat. I believe this is a critical step to ensure that the 'micro-topographic patterns' that are described in the manuscript are realistic.

Other concerns:

1. The vertical accuracy of the lidar data was not provided, or at least I could not readily find it within the manuscript. The horizontal resolution of the various lidar data was provided but this is less important than the vertical accuracy of the lidar when determining elevation or if the lidar data is accurate enough to determine local changes in elevation.

2. The impacts of benthic micro-algae on sediment cohesiveness was discussed as one factor in altering flowpaths and generating micro-topographic features. Can the NDVI data be used to determine or estimate benthic microalgal cover and if there is a correlation between high benthic microalgae cover and sediment stability? It is unclear given what has been provided in the methods section.

3. Although frequency in tidal immersion clearly is important in shaping local topography, if the tidal mudflat regions of study are exposed to high boat traffic, presumably boat wakes may alter the hydrodynamic interactions at these shallow water depths. Some discussion or analysis should be conducted to address the influence of wave activity on these shallow mudflat regions and if they influence these state-change dynamics.

4. Numerical 'inclined plane' model: This seems like a rather basic analysis to describe a very complicated flow scenario that also includes feedbacks with sediment type, sediment cohesiveness, and other physical factors. Although I do not have major concerns with the use of this model, there is a large body of work on river channel geomorphology that has studied these flow networks. I feel a greater attention to the limitations of this model, and other factors controlling flow dynamics in relation to topography need to be described either in the introduction or discussion section.

5. Within the introduction describing other stage-driven transitions (line 282): I feel like some specific examples would be warranted here with more description.

Reviewer #2 (Remarks to the Author):

I enjoyed reading the paper on early indicators of tidal ecosystem shifts, and like that the authors worked hard to provide robust proof that these links exist in field observations because usually the only evidence that is provided is from numerical models with poor ground truthing. I found the introduction and discussion generally thoughtful and well referenced.

A couple of major points:

It is not clear to me whether the general elevation and slope of the intertidal bathymetry or the existence of microtopography (as argued here) are the actually precursor to vegetation taking over. In Swales et al., "Mangrove-forest evolution in a sediment-rich estuarine system: opportunists or agents of geomorphic change?" <https://doi.org/10.1002/esp.3759>, he argues that the intertidal morphology reaches an appropriate elevation and then the mangroves expand quickly seaward. I don't think the existence of microtopography plays a role at all. Microtopography does appear at this site from time to time, but then a storm comes and reworks it, so it is a temporary feature. So I am not convinced that micro-topography is the controlling trigger of the stable state.

Another difficulty I have is on how microtopography is quantified. Where I live, densely vegetation salt marsh and mangrove areas exist on the upper intertidal, and I agree that it would be easy to omit these areas using NDVI ratios. However, in the intertidal region below this, seagrass coverage is really common, with coverage ranging from 5% to 100%. These seagrasses patches grow outward from a centre, making round patches everywhere in the estuary. How do you omit these? Or maybe they are part of the microtopography?

I think I agree with the outcome of the simple flow-routing model as a potential explanation of the link between the formation of microtopography and the slope. However, there are many things neglected in this approach with should be acknowledged (pressure gradients, friction). I am pretty sure the coauthors have the ability to test this with momentum-conserving (and mass-conserving) model.

Nevertheless, I found the paper thought provoking and inspiring, and I think it should be published.

Minor comments.

L57 I am sure these papers are not about internal waves. Do you mean wind waves within the estuary?

L68 the implication here is that environment managers manage for profit...maybe profit is the wrong word. I don't know many managers who's main goal is to improve profitability. Maybe instead say "to improve ecosystem services"/

120 what does de-embanked mean?

I really don't see the need for making a complex sounding parameter of "probability of inundation skipped". Why not say "probably of emergence" (emerge means to become exposed and is often used to mean the opposite of immerge).

178 a nice explanation of the dependence on slope. I through that "droplets" maybe a poorly chosen word as a model like this does not model droplets. Droplets are controlled largely by surface tension. I think a better word would be water parcels.

L206 what causes the flattening?

P269 It is odd to select this as examples .. there are many human activities that affect morphology, like causeways, roading, seawalls, reduced (or increased) sediment supply ...why the focus on dredging

I am not so keen on the last 2 paragraphs of the discussion. They do not really add a lot of insight. It would be better to provide recommendations on data needed to explore their hypothesis more thoroughly.

Reviewer #3 (Remarks to the Author):

This manuscript presents a novel way of forecasting the establishment of vegetation in tidal flats, and thus its transition to a marsh, using the presence and intensity of micro-topography within the tidal flat. By linking the slope of the tidal flat to the development of micro-topography, the authors propose simple geomorphic metrics to describe the state change. This is supported by a neat process-based conceptual picture.

The topic is certainly interesting and relevant, and the scale and depth of the data analyzed fits the scope of the conclusions. However, there are several inconsistencies in the way the data is presented and shown in the figures that undermine authors' conclusions. Therefore, I cannot recommend publication until those are properly resolved or clarified.

Major points:

1. A key conclusion is that the intensity (or degree of development) of micro-topography is crucial for the development of relatively low-elevation marshes. As shown in Fig.3, this is supported by the dependency of the probability of new vegetation on the two explanatory variables: (1) the intensity of micro-topography and (2) the percentage of tidal inundations skipped. However, as the authors state (and show in Fig.S3), they are not independent variables and a more intense micro-topography correlates with higher elevation. Since it is well known that higher elevation leads to new vegetation, that would contradict the relevance of Fig.3 and authors' conclusion.
2. The range of the micro-topography metric (SD of NIR band) is different in all the figures: in Fig.S3 it goes from ~4 to ~7; in Fig.4A, from ~3.5 to ~4.5; and in Fig.3, from 1 to 50. This is relevant because, as shown in Fig.3, the range above 7 is the one better predicting vegetation establishment at lower elevations, but this range is excluded from the comparison with slope (Fig.4) and tidal inundations skipped (Fig.S3). Without this information, there is no way to verify the actual relation between tidal flat slope and marsh establishment, or properly check the range where the explanatory variables are actually independent (see point above).
3. The most important explanatory variable, the micro-topography metric (defined as the standard deviation of the NIR band), is never calibrated. There is no way to know what is the lower bound for the absence of micro-topography or how it increases with the actual micro-topography relief. For example, in Fig.4A the authors imply that a micro-topography metric less than 3.5 represents the absence of micro-topography (for large enough slopes). However, this is not obvious from Fig.3, where there are points with a micro-topography metric as low as 1.

Minor suggestions:

4. Can you add another plot with the frequency of each pair (x,y) in Fig.3 to better illustrate the relation between the two explanatory variables?

5. In Fig. 4A, either add the standard deviation of the micro-topography metric for each slope bin or create a scatter plot (or 2D histogram) with all data, to better understand the relation with local slope.

6. In Fig.S2, add a plot as panel C but vs. micro-topography metric (perhaps for tidal inundations skipped < 60%) to better understand the average relation between new vegetation and micro-topography without considering local elevation.

7. Line 314 - It is not clear what do you mean by "inverted".

Dear anonymous reviewers,

We are pleased to submit to you our point-by-point response to your comments and concerns on our manuscript entitled 'Early indicators of tidal ecosystem shifts are robust across estuaries'.

On behalf of all co-authors, I would like to thank you for your critical but extremely constructive and supportive feedback on our manuscript. I hope you will find that the revisions we have made to the manuscript have made the narrative of our study clearer and our arguments more convincing on the whole. There have been significant revisions to our results, methods, and discussion, as well as a number of new supplementary figures, and revisions to two of the main manuscript figures. The full details of these revisions are structured in a point-by-point response to each reviewer's concerns below.

Among the most notable revisions are:

1. A calibration of our 'intensity of micro-topography' metric where we compare our photo technique to high spatial resolution DEM model gathered via terrestrial lidar,
2. A complete rework of our numerical model using a mass- and momentum-conserving flow model,
3. And a deeper analysis of the relationship between micro-topographic pattern formation and both the slope and elevation of the tidal flat.

We hope you find that these changes have significantly improved the quality of our original manuscript. Of course, we would be happy to make further changes as you see fit.

Thank you for your time and consideration,

Greg Fivash

Color Key:

In response letter:

Black text: Reviewer/editor feedback (Numbered, i.e., 1.1),

Blue: Author responses (Numbered i.e., R1.1)

In quoted manuscript passages (Line numbers, i.e., L50-51):

~~Red strikethrough~~: Removed text, Green: New text, Black: Unchanged text

Lastly, to aid the readability of quoted passages,

In-text references to literature are in **BLUE**, and

In-text references to figures or supplementary materials are in **ORANGE**

Dear Mr Fivash,

Thank you again for submitting your manuscript "Early indicators of tidal ecosystem shifts are robust across estuaries" to Nature Communications. We have now received reports from three referees, based on which we have decided to invite a major revision of the manuscript.

As you will see from the reports copied below, the reviewers find the work of high interest but raise some important concerns. We find that these concerns limit the strength of the study, and therefore we ask you to address them with additional work, including:

1. the analyses recommended by Reviewer 1 and Reviewer 3 to support the micro-topography metrics

This has been performed. See response R1.1

2. and the analyses recommended by Reviewer 2 to test for some factors not included in the model.

This has been performed. See response R1.5

If you feel that you are able to comprehensively address the reviewers' concerns, please provide a point-by-point response to these comments along with your revision. Please show all changes in the manuscript text file with track changes or colour highlighting. If you are unable to address specific reviewer requests or find any points invalid, please explain why in the point-by-point response.

REVIEWER COMMENTS

Reviewer #1 (Remarks to the Author):

This manuscript describes a study that primarily utilizes remote sensing data to determine how the ecological transition of intertidal mudflats to vegetated marshland can be detected through early warning first by the development of micro-topographic patterning. The researchers suggest that this early indicator of a transition occurs only on very shallow sloped tidal flats with a slope <0.5 degrees and occurs due to the impacts of a draining surface at elevations just above the neap high tide water level. A basic numerical model was used to show that at flatter surfaces with slopes near zero that pooling of water droplets occur at local elevation minima, while at larger slopes (>0.5 degrees), they flow down the incline along a direct path and tend not to generate flowpaths that link up to circumvent minor variations in the surface elevation.

Overall, I found this to be a very fascinating study that clearly has large implications for the fate of coastal mudflats and marshes in the face of sea level rise and other anthropogenic threats (including deepening of shipping lanes through dredging). The authors did a great job within the introduction of describing the relevant theory and benefits of determining early warning signs of ecosystem shifts and how it applies to intertidal mudflat/marsh systems. Their conclusions seem robust across 3 large estuary systems in Europe, and I believe this could be a seminal paper that will spur new research in to topographic influence on state-change dynamics. With the ever-increasing accuracy of digital elevations from lidar, GPS systems, and drone imagery, this will likely be an increasing area of research. The manuscript is very well written with a clear logical progression, and good use of figures and supplementary material.

We could not have hoped for a better reception. The authors feel honored to have been able to produce work that meets this standard.

1.1

I do have one major concern and a few more minor ones that I feel need to be addressed:

The one major criticism I have is the application of how micro-topography was quantified and used broadly within this study. The authors chose to measure the presence and intensity of micro-topography through orthophotos that had a horizontal resolution of 25-50 cm. The magnitude or 'intensity' of micro-topography on a tidal flat was then calculated using the standard deviation in the NIR band within a 10m x 10m frame. No micro-topographic elevations were in fact measured at all. This may be a reasonable albeit 'novel' way to estimate micro-topography, but there was no mention within the methods of validating this method in any way. The entirety of their results rely upon the assumption that the standard deviation of the orthophotos are a good predictor of topography. Although I think this is a useful way to quantify micro-topography features using coarse data, it should be validated. Was any high resolution topography quantified over these small scales to compare to the standard deviations in the NIR band? This perhaps can be done using structure-from-motion analysis of drone flights to obtain higher resolutions, or more detailed gps surveys over smaller sections of a tidal flat. I believe this is a critical step to ensure that the 'micro-topographic patterns' that are described in the manuscript are realistic.

R1.1

It's obvious this validation needs to be performed and we apologize that this was not included in the original manuscript. In order to address this concern, we have added a new section to the methods titled: *Calibration of the micro-topographic intensity metric* and included a new supplementary figure: Supplementary Fig. 7).

In this new section we compare our 'intensity of micro-topography' metric with the results of terrestrial lidar surveys with higher resolution elevation measurements at a number of sites in the Western Scheldt. All the details can be found in that section as follows:

L 459:

Calibration of the micro-topographic intensity metric

In order to test the reliability of using images to characterize geomorphological features on the tidal flat, we performed a calibration of our 'intensity of micro-topography (SD of NIR band)' metric by comparing it to high resolution terrestrial lidar data. Between 2020 and 2022, we performed repeated terrestrial lidar surveys (RIEGL VZ-400i, RIEGL Laser MeasurementSystems GmbH, Horn, Austria) of three tidal flats in the Western Scheldt harboring micro-topographic patterns (Hoofdplaat: 51.373, 3.675; Zuidgors: 51.391, 3.855; and Baarland: 51.393, 3.866). The elevation point clouds produced by these surveys were converted to 25cm resolution rasters matching the dimensions of the publically available orthophotomosaics. We then calculated the local slope of the tidal flat using the terrain function available in the R 'raster' library (Hijmans 2020)⁶⁷. We chose to measure variation in slope rather than the variation in elevation so that gradually sloping surfaces were not falsely labelled as having micro-topography. The

orthophoto mosaics taken of the Western Scheldt in 2020, and 2022 were paired with lidar surveys that occurred nearest in time, all at least within six months of each other. The standard deviation of the (1) local slope and of the (2) NIR band intensity were measured in 10 x 10m grids over the regional extent of the terrestrial survey. Then, these two metrics were compared using a linear regression after log-transforming both variables to normalize the distribution of the data. A strong correlation was found between these two variables, indicating that our approach for detecting micro-topography from NIR photographs is reliable when elevation data of suitable resolution is not available (Supplementary Fig. 7, $F_{1,1176} = 2966$, $R^2 = 0.72$, $p < 0.001$).

Supplementary Figure 7. A calibration of the technique of using photos to measure micro-topography, compared with elevation measurements using terrestrial lidar. In panels A & B, the elevation and slope rasters produced by aerial lidar is compared with the higher resolution product of a terrestrial lidar survey (RIEGL VZ-400i, RIEGL Laser Measurement Systems GmbH, Horn, Austria). While the large-scale patterns in elevation change are captured equally in both methods (panel A), it is clear that the micro-topographic patterning is mostly averaged out in the coarser aerial lidar surveys (panel B). Terrestrial surveys are however not available for the entire estuary. Fortunately, the micro-topographic patterns detected in terrestrial surveys (panel B) are also visible in the estuary-wide NIR-band orthophotomosaics (panel C). In panel D, we compare the standard deviation of the slope captured by terrestrial lidar with the standard deviation of nir-band photos, measured over a 10 x 10m sampling grid, and a strong correlation between the two metrics can be seen.

1.2

Other concerns:

The vertical accuracy of the lidar data was not provided, or at least I could not readily find it within the manuscript. The horizontal resolution of the various lidar data was provided but this is less important than the vertical accuracy of the lidar when determining elevation or if the lidar data is accurate enough to determine local changes in elevation.

R1.2

The vertical accuracy of the lidar data provided for the three estuaries had a **1 cm** vertical resolution.

This information has been added to the supplementary appendix 1 alongside information about the horizontal resolution of lidar data that was originally included, in the following passages:

L948, Western Scheldt: ‘The elevation data is available in annual time series for the period from 2004 to 2020, at 2 – 5 m horizontal resolution and 1 cm vertical resolution’

L981, Elbe: ‘In this case aerial lidar data (1 m horizontal, 1 cm vertical resolution)’

L1019, Humber: ‘These data included aerial lidar data (1 m horizontal resolution and 1 cm vertical resolution,’

1.3

The impacts of benthic micro-algae on sediment cohesiveness was discussed as one factor in altering flowpaths and generating micro-topographic features. Can the NDVI data be used to determine or estimate benthic microalgal cover and if there is a correlation between high benthic microalgal cover and sediment stability? It is unclear given what has been provided in the methods section.

R1.3

In principle benthic algae can be detected in NDVI data and the presence of benthic algae does generally indicate stable sediment. However, in practice using algae as a proxy for sediment stability can be troublesome since their presence and density fluctuates seasonally, independent of changes in sediment stability. Algae also migrate with the tide, which introduces another source of variability. The question as a whole is a bit outside the scope of this study and in my own estimation it is beyond the capabilities of this dataset to test. This is because we have access to only semi-annual orthophotos that were not always taken in a standard month. This makes controlling for the strong seasonal and tidal affects very challenging. In addition to this, we also lack large-scale measurements of sediment strength

to test those data against. More frequently collected drone imagery of a smaller area would be more appropriate to address this question. For the interest of the reviewer and authors, such studies are on-going in our department, but will be part of a separate manuscript.

Note also that cohesive sediment does not require benthic algae to become more cohesive. The build-up of a cohesive surface layer also occurs as a physical consequence of dewatering. Further investigation into the role this may play is also on-going in our department.

1.4

Although frequency in tidal immersion clearly is important in shaping local topography, if the tidal mudflat regions of study are exposed to high boat traffic, presumably boat wakes may alter the hydrodynamic interactions at these shallow water depths. Some discussion or analysis should be conducted to address the influence of wave activity on these shallow mudflat regions and if they influence these state-change dynamics.

R1.4

Studies such as Schroevers *et al.* (2011) clearly show that boat wakes play an important role in eroding the shorelines of estuaries like the Western Scheldt. However, it is not clear that the erosion of these shorelines has any relationship with the deposition of sediment on upper intertidal flats, which is a central focus of this study. Given the quality of data and complexity of analyses that would be required to properly investigate the question posed by Reviewer 1 (long term wave data at many key points along the estuary), the authors deem this question outside of the scope of this largely geospatial study. We see also that the sites of vegetative expansion are generally getting higher and wider over time (true generally over the estuary as seen in Figure 2, and on specific sites of expansion as in Supplementary Fig. 4). This indicates that erosive forces, including those caused by boat wakes, are not able to out-pace the forces driving accretion.

We have attempted to address this point in the text by adding a passage in the introduction directing readers to the work of Schroevers *et al.* on boat wakes in estuaries that we hope you will find satisfactory.

New text (L53-58):

What results is a deeper and narrower estuary, which magnifies the tidal amplitude and current strength (~~van Maren *et al.* 2015, Stark *et al.* 2017, van Dijk *et al.* 2021~~)^{8,12,13}, while also curtailing ~~internal-wave generation~~ wind-driven wave generation within the estuary, due the shortening fetch length (~~Friedrichs 2011, Green & Coco 2014, de Vet *et al.* 2017~~)^{7,14,15}, offset to some degree by the increasingly role of erosive ship wakes (~~Schroevers *et al.* 2011~~)¹⁶.

Citation added:

16. Schroevers M., Huisman B.J.A., van der Wal M., Terwindt J. Measuring ship induced waves and currents on a tidal flat in the Western Scheldt estuary. *IEEE/OES 10th Current, Waves and Turbulence Measurements (CWTM)* 123-129 (2011). <https://doi.org/10.1109/CWTM.2011.5759539>.

1.5

Numerical 'inclined plane' model: This seems like a rather basic analysis to describe a very complicated flow scenario that also includes feedbacks with sediment type, sediment cohesiveness, and other physical factors. Although I do not have major concerns with the use of this model, there is a large body of work on river channel geomorphology that has studied these flow networks. I feel a greater attention to the limitations of this model, and other factors controlling flow dynamics in relation to topography need to be described either in the introduction or discussion section.

R1.5

In order to address the comments made by Reviewer 1 here and also encompass suggestions made by Reviewer 2 (in comment 2.3), we have made a number of significant changes to the numerical model and the discussion of this topic in the manuscript to improve the aspect of the manuscript that deals with the role of slope in the formation of these patterns.

To begin, we decided to significantly revise our numerical model as suggested by Reviewer 2 to increase its realism. The simple pixel flow algorithm found in the earlier version has been replaced with a surface flow model described in Weerman *et al.* 2010. This model remains simple, but includes conservation of mass and momentum, as requested by reviewer 2.

This change has not affected the main conclusions drawn from the original model. It remains true that as we tilt the inclined plane to be steeper, flow patterns eventually transition from a state of more concentrated flow down specific pathways toward a state where the water flows down the plane as a sheet, distributed fairly equally over the plane.

However, as you can see in our new figure on this topic (Figure 4), the conservation of mass does change our results when the plane is perfectly flat. In the original model, the paths taken by water parcels were followed in separate simulations so they could not interact to form pools. In this new model, it is now possible, and we see that the concentration of water no longer drops when the plane is nearly flat. Ultimately, we think this new model creates a more accurate representation of reality. Draining surface water really does form pools on low slope tidal flats during low tide. However, this means that we can no longer explain the measured decrease in micro-topographic intensity in nearly flat areas (< 0.05 degrees, see Figure 4) in the geospatial data to be a consequence of the slope dynamics alone, as we had done so in the earlier version.

It is the opinion of the authors that in reality, the lack of sediment dynamics in very flat landward tidal flats must also be considered to explain this phenomenon. But without adding even more extraneous data to the paper it is difficult to broach this topic here. Thus, we have chosen to note this difference between model expectations and reality and discuss it briefly (see changes to the discussion below). We hope that the reviewers (especially reviewer 2) will find these changes provide an acceptable solution to their concerns.

Below, we link the complete set of changes related to this comment:

New methods section:

3. Numerical ‘inclined plane’ model (MOVED TO APPENDIX 5)

Lastly, we describe a simple numerical model that provides an explanation for the association between low-slope tidal flats and the appearance of micro-topographic drainage patterns. In these simulations, the flow pathway of a water parcel originating at some point on an inclined plane was followed until it reaches the lower boundary of the plane or settled in a local elevation minimum. A small amount of surface variability (SD: 2 mm, assuming a cell size of 1 m²) was added to the plane in the form of auto-correlated noise (created through the fractal folding method described in Saupe 1988). Water parcels travelled down stream by moving iteratively from their origin cell to the neighboring cell with the largest elevation gradient (G). This is described in Equation 2, where z , x , and y , represent the vertical and two horizontal coordinates of the parcel’s current cell (z_{cell}) and its eight neighbors ($z_{neighbor}$):

$$G = \frac{(z_{cell} - z_{neighbor})}{\sqrt{(x_{cell} - x_{neighbor})^2 + (y_{cell} - y_{neighbor})^2}} \quad \text{Equation 2}$$

The model’s purpose is to demonstrate how the concentration of draining surface water is affected by the slope of the underlying surface when minor local slopes caused by randomly distributed small topographic irregularities are also present. In a series of repeated simulations, the inclination of the plane was gradually increased. For each inclination in the series, the flow pathway was simulated for one parcel originating at each cell in the plane. A record was kept of the number of parcels that travelled through each cell at a given inclination. The intensity of flow concentration over the plane was then quantified by measuring the standard deviation of the number of parcels that flowed through each cell. This value was then compared between trials with different underlying slopes. These trials used a 256 x 256 cell grid with a repeating boundary condition at the sides of the plane, and an impermeable boundary at the top. All geospatial analyses and numerical modelling were performed in R (R Core Team 2020).

3. ‘Inclined plane’ numerical model

Here, we describe a simple numerical model that provides one possible explanation for the association between low-slope tidal flats and the appearance of micro-topographic drainage patterns. As our focus is on understanding the drainage patterns, we do not simulate full tidal cycles. Instead, we simulate the drainage of a thin layer of water, which can be considered as the drainage during ebb tide at the moment the tidal flat starts to become dry after a tidal inundation. In these simulations, we follow the flow of water over an inclined plane to see how patterns in flow concentration change depending on the slope of the plane. A small amount of surface variability (SD: 2 mm, assuming a cell size of 1 m²) is added to the plane in the form of auto-correlated noise (created through the fractal-folding method described in Saupe⁷⁰ 1988). The model’s purpose is to demonstrate how the concentration of draining surface water is affected

by the slope of the underlying surface when minor local slopes caused by randomly distributed small topographic irregularities are also present.

The flow of water over the inclined plane is simulated using the flow equation described in Weerman *et al.* 2010⁴⁶, expanded into two dimensions (Equation 2). With this simple formulation, the velocity and direction of flow are determined by a combination of the water level and local slope of the plane, while maintaining conservation of mass and momentum.

$$\frac{dW}{dt} = \frac{d}{dxy} [KW \frac{d(W+S)}{dxy}] \quad \text{Equation 2}$$

Here, W and S represent the local height (cm) of the overlying water layer (W) and the surface of the plane (S), respectively. Hydraulic conductivity (K), originally a function of water depth, is here set to a constant value equal to Weerman's⁴⁶ original minimum value ($K = 1$) due to the very shallow water levels explored in this model. The simulation begins with a small initial quantity of water spread evenly over the plane (W_i). To ensure that the model consistently simulated the flow dynamics of very shallow surface water that would be re-directed by the topography of the tidal flat, the depth of this initial water layer (W_i) was set to equal 10% of the standard deviation of the autocorrelated noise. The water then flows over the plane until the average changing water level across the plane is very small ($\leq 0.3\%$ of W_i). In a series of repeated simulations ($n = 100$), the inclination of the plane was gradually increased over a range between 0 and 2 degrees. For each inclination in the series, the water level in each cell was recorded at each time step. In order to measure the extent to which the water tended to concentrate into specific regions or flow pathways, we calculated the average water height in each cell over the simulation and then measured the standard deviation of the average water height across the grid. This value, which we called the 'intensity of flow concentration' (see Figure 4b) was then compared between model runs with different underlying slopes.

These simulations used a 512 x 512 cell grid with repeating boundary conditions. Water parcels moving over the top-bottom boundary (returning to the top of the plane after flowing down it) continued to flow as if they had experienced a continuous gradient, via the following formulation:

$$S_{nx+1} = S_1 - (S_{max} + \frac{S_{max}}{nx}) \quad \text{Equation 3a}$$

$$S_0 = S_{nx} + (S_{max} + \frac{S_{max}}{nx}) \quad \text{Equation 3b}$$

Here, S_{max} indicates the maximum height of the inclined plane and nx is the length of the plane. S_0 and S_{nx+1} represent neighboring cells that are beyond the extent of the grid (S_0 is before the first index, S_{nx+1} is after the last index).

New results section:

L174:

'Micro-topography forms on gently sloping upper intertidal flats

By using lidar data and false color images in combination to detect very small-scale topographic features, we determined that micro-topographic patterns are found predominantly on tidal flats with very shallow slopes, peaking in intensity at around 0.1 degrees (Supplementary Fig. 3d). As with vegetation establishment, the intensity of tidal flat patterning is also consistently stronger in areas that experience more frequent relief from tidal inundation. (Supplementary Fig. 3c). However, no matter the elevation, micro-topographic patterns tend not to occur on tidal flats with a slope greater than 0.3 degrees (see

Supplementary Fig. 3a). ~~Micro-topographic patterns occur much more frequently on tidal flats with a slope less than ~0.5 degrees. Like vegetation, these patterns also seem to occur preferentially above neap high water and decrease sharply in frequency in areas that experience less frequent relief from tidal inundation (Figure S3). Thus, patterns can be expected to appear most often on accreting, convex upper intertidal flats, which feature flat anterior sections high in the tidal frame (see Supplementary Fig. 4 for an example).~~

~~*Modelling indicates that micro-topography forms as a consequence of the concentration of surface water*~~

~~Simple simulations give insight into one explanation of why micro-topographic features occur in low-slope environments. In these simulations the direction of flow is determined locally by the direction of the steepest slope. This mechanism ultimately causes draining surface water to concentrate into united flow pathways in low slope environments. Here, the local slope depends on the combination of two factors: (1) the underlying slope caused by the inclination of the plane, and (2) the local slopes caused by small irregular features imposed on the landscape. When the slope of the underlying plane is zero, these small features tend to cause pooling of water at local elevation minima (i.e., see 0.006 degree slope in Figure 4c). When the underlying slope is too severe, the slopes caused by small landscape features are not large enough to cause flow deviation in a direction other than that directed by the inclined plane (i.e., above 0.15 degrees in Figure 4c). However, when the underlying slope drives water parcels to overcome minor boundaries in the landscape and circumvent larger ones in order to ‘drain’ out of the inclined plane, flow pathways tend to link up with others (i.e., between 0.025 to 0.1 degrees in Figure 4c). This concentrates the flow of water over specific pathways of least resistance in the landscape.~~

~~*Micro-topography may arise from patterns in draining surface water*~~

Simple simulations modelling the shallow flow of draining surface water over an inclined plane give insight into one explanation of why micro-topographic features occur in low-slope environments. In these simulations, the direction of flow over the plane is determined predominantly by the local direction of the steepest slope. Here, the local slope depends on the combination of two factors: (1) the underlying slope caused by the inclination of the plane, and (2) the local slopes caused by small irregular features imposed on the bed elevation. When the slope of the underlying plane is near zero, the small features in the bed elevation tend to cause water to pool at local elevation minima (Figure 4). ~~(i.e., see 0.006 degree slope in Figure 4c), which does not facilitate the concentration of flow along specific pathways).~~ As the tilt of the underlying slope increases, ~~droplets~~ water parcels begin to overcome small raised features in the bed elevation and circumvent larger features in order to ‘drain’ out of the inclined plane, leading to flow pathways that tend to link-up with others. ~~(i.e., between 0.025 to 0.1 degrees in Figure 4c).~~ This concentrates the flow of water over specific pathways of least resistance in the landscape. But if the underlying slope becomes too severe, the local slopes caused by small features in the bed elevation will eventually cease to be large enough to cause flow deviation in a direction other than that directed by the larger overall slope of the inclined plane. ~~(i.e., above 0.15 degrees in Figure 4c).~~ The result is that, depending on the magnitude of the irregular bed features, a certain tilt in the landscape will cause a shift from flow concentration in pools and drainage pathways to sheet flow where a similar amount of surface water flows down the plane everywhere. When we set the standard deviation of the irregular bed features to 2.5 cm (assuming a cell size of 1 m²), the dynamics of our simulation appear to reflect the distribution of micro-topographic patterns in the geospatial data (Figure 4a & 4b). Note that this model cannot explain the decrease in micro-topographic intensity at near-zero slopes observed in the geospatial data (see Figure 4a).’

New discussion passages:

L258:

‘...The result of the estuary-scale analysis in this study shows a clear correlation between the slope of the tidal flat and the intensity of micro-topographic patterning (Figure 4a). The results of our ‘inclined plane’ model suggest that the formation of micro-topographic patterns may be related to the formation of pools and the concentrating flow pathways of draining surface water in nearly flat environments (Figure 4). While this model provides a robust explanation for why patterns do not form in steeper areas, it cannot provide a complete explanation of the conditions required for patterns to form. It cannot, for instance, explain the decrease in micro-topographic intensity in near-zero slope environments that is measured in the geospatial data. We have also found that micro-topographic patterning is more intense when the tidal flat is frequently above water during high tide (Figure 4b). This suggests that the low tide period itself may play an important role in the amplification of these patterns, perhaps by exposing the sediment to prolonged periods of drying. It is also worth considering that the widening and heightening upper intertidal flat will also experience a reduced wave climate, which will reduce sediment turn-over (Friedrichs 2011)¹⁴. Each of these various environmental characteristics in combination are likely important to the formation of permanent micro-topographic patterns. The major weakness in our understanding of these patterns up to this point is that has so far been guided exclusively by correlative evidence.

We still cannot yet provide a complete explanation of how and why these high, low-slope tidal flat environments tend to produce micro-topographic bed level variation, but we can provide some direction for further inquiry: ...’

New figure(s):

Figure 4. Comparison of model results with the distribution of micro-topographic patterns in the geospatial data. Panel A displays the preferential occurrence of micro-topographic patterns on low-slope tidal flats considering all the three studied estuaries ~~in the Western Scheldt~~. Here, points represent the mean value of each binned group, and error bars show standard error. Panel B displays flow concentration predictions produced by the ‘inclined plane’ numerical model over planes of varying slope. Points show the average result across 100 simulations at each inclination, each using a unique randomly generated surface of autocorrelated topographic noise. Panel C visualizes how the distribution of surface water varies as a consequence of the underlying slope. Here the outcome of six simulations are displayed side-by-side, each with the same pattern of topographic noise, but with a steepening overall inclination from left to right. Note that in each of these panels, the plane inclines from top to bottom, so that the predominant flow direction is from top to bottom. We compare the spectrum of simulated outcomes, ranging between a landscape of pooling water (C, left) and ubiquitous ‘sheet’ flow (C, right) to geospatial data in the two lowest panels. Here, at the tidal flats near Hooftplaat, in the Western Scheldt, micro-topographic patterning containing pooling surface water abruptly ends (D) when the slope of the tidal flat begins to steepen (E, 2m res.).

Supplementary Figure 3. The effects of tidal flat slope and position in the intertidal on micro-topographic intensity are displayed in four panels, considering all three studied estuaries. In Panel A, the interactive effects of tidal position and slope on micro-topographic intensity are explored over a 2D field. Here darker colors represent on average more intense micro-topographic patterning. Panel B shows the number of observations at each cell in the field (cells supported by less than 30 observations have been removed). Note the strong correlation between high position in the intertidal and low slope tidal flats, indicated by the distribution of observations across this frame. Panels C and D display the effects of tidal inundation (C) and slope (D) independently. Here, points represent the average of a binned group and error bars show standard error. These panels show that micro-topography is stronger when higher in the tidal frame on very shallow slopes. However, panel A shows that micro-topography only occurs in the upper intertidal within low slope regions (slopes less than 0.2 - 0.3 degrees), indicating that slope may be the predominant precursor of pattern formation.

1.6

5. Within the introduction describing other stage-driven transitions (line 282): I feel like some specific examples would be warranted here with more description.

R1.6

This comment refers to the passage:

‘Apart from tidal flat-marsh transitions, it is likely that similar stage-driven transitions occur in other ecosystems that have been described to follow alternative stable state dynamics (Schröder *et al.* 2005, van der Heide *et al.* 2007, Heffernan, 2008, Wang & Temmerman 2013, Kéfi *et al.* 2016, Wang *et al.* 2016).’

As requested, we’ve expanded on the suggestion in this line with some further discussion, as follows:

L360:

The development of biogeomorphic landscapes from bare to vegetated states has been discussed at length under two alternative frameworks: (1) community succession (~~Corenblit *et al.* 2007, Han *et al.* 2022~~)^{57,58}, and (2) alternative stable state theory (~~Schröder *et al.* 2005, van der Heide *et al.* 2007, Heffernan 2008, Wang & Temmerman 2013, Kéfi *et al.* 2016, Wang *et al.* 2016~~)^{21,22,59,60,61}. In this example we see that a fusion of these two concepts best explains the transition. Here, a steady environmental change (in this case constant sediment accretion), eventually reaches a threshold at which a cascade of additional changes begins to take place. These secondary changes could be driven by physical mechanisms (like prolonged sediment de-watering) or by pioneering (micro-)organisms that appear prior to the vegetation transition. A similar case could be made for arid ecosystems where the development of a biologically active soil crust, which depend on adequate rainfall and limited erosion, is a critical precursor to the invasion of higher plants (~~Bowker 2007~~)⁶². Apart from these examples ~~tidal flat-marsh transitions~~, it is likely that similar stage-driven transitions occur in other ecosystems that have been described to follow alternative stable state dynamics.

Added references:

Corenblit D, Tabacchi E, Steiger J, Gurnell AM. 2007. Reciprocal interactions and adjustments between fluvial landforms and vegetation dynamics in river corridors: A review of complementary approaches. *Earth-Science Reviews* 84(1–2): 56-86. doi:10.1016/j.earscirev.2007.05.004.

Han M, Brierley G, Pan B, Geng H, Shi Y. 2022. An approach to evaluate the dominant river biogeomorphic succession phase at the reach-scale. *CATENA* 217: 106455. doi:10.1016/j.catena.2022.106455.

Bowker MA. 2007. Biological soil crust rehabilitation in theory and practice: an underexploited opportunity. *Restoration Ecology* 15:13–23. doi:10.1111/j.1526-100X.2006.00185.x.

Reviewer #2

(Remarks to the Author):

I enjoyed reading the paper on early indicators of tidal ecosystem shifts, and like that the authors worked hard to provide robust proof that these links exist in field observations because usually the only evidence that is provided is from numerical models with poor ground truthing. I found the introduction and discussion generally thoughtful and well referenced.

Thank you

A couple of **major points:**

2.1

It is not clear to me whether the general elevation and slope of the intertidal bathymetry or the existence of microtopography (as argued here) are the actually precursor to vegetation taking over. In Swales et al., "Mangrove-forest evolution in a sediment-rich estuarine system: opportunists or agents of geomorphic change?" <https://doi.org/10.1002/esp.3759>, he argues that the intertidal morphology reaches an appropriate elevation and then the mangroves expand quickly seaward. I don't think the existence of microtopography plays a role at all. Microtopography does appear at this site from time to time, but then a storm comes and reworks it, so it is a temporary feature. So I am not convinced that micro-topography is the controlling trigger of the stable state.

R2.1

No doubt we would need a separate study to determine any role micro-topography might play in mangrove establishment and it is possible that we would find them to not play a role in that scenario. It is also true that micro-topography can be transient, and in those cases we agree that it is very unlikely that they would drive vegetation expansion. However, in the areas detailed in this study, micro-topography was generally already present and persistent a minimum of one to three years prior to vegetation expansion. Experimental evidence supporting the facilitative role of raised surfaces in micro-topographic patterns for European salt marsh pioneers can be found in a number of earlier publications (Mossman et al. 2020, Fivash et al. 2020 & 2021, van der Vijssel et al. 2021).

Within this study, Figure 3 provides very strong evidence of the complementary facilitative effect micro-topography has alongside elevation for marsh expansion in these estuaries. If micro-topography played no role, we would expect to see a horizontal boundary between the unvegetated and vegetated regions of the figure. Instead, we see a clear diagonal boundary, indicating that with stronger micro-topographic patterning, vegetation can establish lower in the intertidal. We've tried to clarify this effect even more so in the new Supplementary Fig. 9, which (at the request of reviewer 3) shows the effects of tidal position

and micro-topography on vegetation establishment in separate panels. We've also changed the color pallet of the original Figure 3 to make it easier to visually distinguish between intermediate values across the gradient between 0 and 100% establishment. We hope that this change will make the diagonal line of the figure clearer to readers at a glance and convince more readers of the importance of micro-topography in vegetation expansion to lower positions in the tidal frame.

Figure 3. Correlation between elevation, micro-topography, and vegetation establishment. The probability of establishment of new vegetation between two consecutive measuring campaigns (1 – 3 yr intervals) is displayed against both (1) the frequency of skipped tidal inundations, and (2) the intensity of local micro-topography. All three sampled estuaries are combined, for all years of data available, to create this figure. Binned groups with $n < 30$ replicate measurements were excluded to eliminate noise in the image. Vegetation expands more frequently at higher position in the tidal frame, and generally cannot establish below ‘neap high water’, when 0 % of tidal inundations are skipped. But areas with strong micro-topographic patterning support the establishment of vegetation at lower intertidal position, closer to that fundamental boundary. ~~Areas with strong micro-topographic patterning support growth of vegetation at lower intertidal position~~ (note diagonal edge of the dark green ‘vegetated’ area, marked by the dashed line). ~~Vegetation still cannot establish when 0 % of tidal inundations are skipped, regardless of the micro-topography.~~

Supplementary Figure 9. An extended version of Figure 3. In addition to the original figure (A), here we also display the total number of observations represented within each cell in the raster (B) and separate the effects of the tidal position (panel C) and the intensity of micro-topography (D) on vegetation establishment into 1D scatterplot figures. The points in panels C and D represent the mean value within each binned group, and the error bars show standard error.

As a final side note to Reviewer 2 (what follows is not a formal revision, just a discussion):

Given the large time-steps used in Swales et al. (1950 - 2006 with 10 year intervals, see Fig. 3), it might well be that intermediate stages in landscape succession were missed. After all, the role of microtopography was also originally overlooked in marsh establishment in our own study system (e.g., see Wang et al. Remote Sens. 2020, 12, 2316; doi:10.3390/rs12142316).

Below we have included some anecdotal evidence showing the presence of micro-topographic patterns in expanding pioneer mangrove areas. These photos are from a presentation provided by Thorsten Balke in 2013, who performed fieldwork with Swales in the Firth of Thames estuary around that time. These images cannot tell us about any effect these patterns may have on pioneer vegetation, or the longevity of the patterns. But their similarity to many of the marsh sites featured in this paper suggests to the authors the possibility that there could be more to the story in mangrove areas. From a physiological perspective, it seems likely that the belowground phenomena that support marsh vegetation in micro-topographic patterns should also aid young mangroves.

2.2

Another difficulty I have is on how microtopography is quantified. Where I live, densely vegetated salt marsh and mangrove areas exist on the upper intertidal, and I agree that it would be easy to omit these areas using NDVI ratios. However, in the intertidal region below this, seagrass coverage is really common, with coverage ranging from 5% to 100%. These seagrasses patches grow outward from a centre, making round patches everywhere in the estuary. How do you omit these? Or maybe they are part of the microtopography?

R 2.2

Seagrass and macroalgae make a mess of conventional NDVI analyses. Fortunately for us, in these three study estuaries both are almost completely absent. In fact, the only seagrass meadow in the Western Scheldt (located near Ritthem at the coordinates: 51.456663, 3.659833) was explicitly omitted from our study to avoid this issue. Of course, benthic microalgae like diatoms also produce a positive signal in NDVI, however it is usually significantly lower than that of vegetation and differentiation is still possible with NDVI alone. Notably, the fibrous epibenthic microalgae *Vaucheria* can occasionally be found in each of these estuaries and it is not easily distinguishable from vegetation with NDVI alone. A certain amount of mis-characterization caused by problems like this one is undoubtedly present in our study. However the occurrence is rare enough to be very unlikely to have effected our main conclusions. Furthermore, our use of NDVI to characterize vegetated areas is already standard practice for studies in this region (see: van Wesenbeeck *et al.* 2008, van Belzen *et al.* 2017, Laengner *et al.* 2019, Wang *et al.* 2020).

Had seagrass and large blooms of benthic macroalgae been a persistent feature in these estuaries, it would have required us to use a more complex characterization method, such as a trained neural network. Note also that a similar tool is already being used by German Federal Waterway authorities in the Elbe estuary, and since it was available, this more accurate vegetation classification was used for the Elbe for this study.

Overcoming this characterization overlap between terrestrial (marsh) and marine (seagrass & algae) photosynthetic species would be key to extending similar analyses to a wider range of study areas in the future, especially in the tropics and sub-tropics.

We have added the following lines to clarify this important point in methods section of the manuscript:

L424:

In analysis of the Elbe estuary, vegetation maps were already available via the German Federal Waterways and Shipping administration (WSV) and were used in place of the above NDVI characterization technique (see Supplementary Methods for details Note that these three temperate European estuaries harbor little to no seagrass or sessile macroalgae, which otherwise would have

complicated the classification of terrestrial vegetation. Benthic microalgae (*e.g.*, diatoms), which are present, produce a positive signal in NDVI as well, however it is usually significantly lower than that of vegetation and differentiation is still possible with NDVI alone. An exception to this is fibrous epibenthic microalgae (*e.g.*, *Vaucheria sp.*), which can occasionally be found in each of these estuaries, and it is not easily distinguishable from vegetation with NDVI alone. A certain amount of mischaracterization caused by problems like this one is undoubtedly present in our study. However, the occurrence is rare enough to be very unlikely to have affected our main conclusions and the use of NDVI to characterize vegetated areas is standard practice for studies in this region (~~van Wesenbeeck et al. 2008, van Belzen et al. 2017, Laengner et al. 2019, Wang et al. 2020~~)^{23,31,65,66}.

Added references:

van Wesenbeeck BK, van de Koppel J, Herman PM, Bertness MD, van der Wal D, Bakker JP, Bouma TJ. 2008. Potential for sudden shifts in transient systems: Distinguishing between local and landscape-scale processes. *Ecosystems* 11: 1133–1141. doi:10.1007/s10021-008-9184-6.

Laengner M, Siteur K, van der Wal D. 2019. Trends in the Seaward Extent of Saltmarshes across Europe from Long-Term Satellite Data. *Remote Sensing*. 11: 1653. doi:10.3390/rs11141653.

2.3

I think I agree with the outcome of the simple flow-routing model as a potential explanation of the link between the formation of microtopography and the slope. However, there are many things neglected in this approach with should be acknowledged (pressure gradients, friction). I am pretty sure the coauthors have the ability to test this with momentum-conserving (and mass-conserving) model.

R2.3

We have done our best to resolve this problem by implementing the flow model with more realistic flow behavior, found in Weerman *et al.* 2010. As requested, this model now considers both conservation of mass and momentum. See comment R1.5 for a full account of the changes.

Nevertheless, I found the paper thought provoking and inspiring, and I think it should be published.

Thank you again

Minor comments.

2.4

L57 I am sure these papers are not about internal waves. Do you mean wind waves within the estuary?

R2.4

Oops! We're sorry for this confusion. Of course, in this passage we meant to refer to wind-driven waves. The original use of 'internal' waves was meant to indicate waves produced *within the estuary*, rather than those generated at sea and entering at the mouth. These waves are reduced in height due to a limited fetch length across the estuary. Clearly, describing these as *internal* waves is an incorrect use of terms and it has been revised accordingly.

See the corrected passage below:

L56:

What results is a deeper and narrower estuary, which magnifies the tidal amplitude and current strength (~~van Maren *et al.* 2015, Stark *et al.* 2017, van Dijk *et al.* 2021~~)^{8,12,13}, while also curtailing ~~internal-wave generation~~ wind-driven wave generation within the estuary, due the shortening fetch length (~~Friedrichs 2011, Green & Coeo 2014, de Vet *et al.* 2017~~)^{7,14,15}

2.5

L68 the implication here is that environment managers manage for profit...maybe profit is the wrong word. I don't know many managers who's main goal is to improve profitability. Maybe instead say "to improve ecosystem services"

R2.5

The original text was meant to evoke the term 'profitable' in the sense of leading to a generally advantageous scenario for managers. We now recognize that the undesirable alternative reading of the text you have pointed out would probably occur to many readers. Therefore, we have rewritten that passage to reflect this intended meaning, as follows:

L71:

Given this trade-off in services, coastal management would benefit greatly from to ability ~~it is profitable for coastal managers~~ to foresee on-coming ecosystem transitions before they begin.

2.6

120 what does de-embanked mean?

R2.6

'De-embankment' refers to a type of management in northern Europe where historical dikes/embankments are broken down to allow a section of agricultural land that was historically 'reclaimed' from the sea to be re-introduced to tidal inundation.

We've added a short definition of this term in parentheses following its first use in the following passage:

L124:

But to date, most of these studies focus on either de-embanked areas (*polders that have been converted back to wetlands by being reintroduced to tidal inundation*), or experiment with artificially created micro-topography in order to develop applications for wetland restoration.

2.7

I really don't see the need for making a complex sounding parameter of "probability of inundation skipped". Why not say "probably of emergence" (emerge means to become exposed and is often used to mean the opposite of immerge).

R2.7

We agree that the term 'probability of inundation skipped' is an unwieldy term and are very open to discussing alternative phrasings.

However, it is crucial to draw a distinction between this metric and that of the more common 'inundation frequency', or its inverse 'emergence frequency'. These high intertidal areas are out of the water most of the time, but can still go under water as often as twice a day, even if it is only for only a few hours. What we measure here is not the amount of time spent in or out of the water, but the likelihood that this submerged period does not occur over a tidal cycle.

In order to address this valid complaint while still drawing a distinction from the more common 'inundation frequency', have replaced the term 'probability of inundation skipped' with 'tidal inundations skipped (%)' wherever it occurs in the text.

Of course, we are very open to any other suggested naming convention the reviewers or editors might have in mind.

See also the new Equation 1 that provides some visual support for our verbal explanation of this term:

L400:

These water level data were used to calculate the frequency of tidal inundations at every intertidal elevation by calculating the number of high tides that reached above a given elevation (*cycles inundated*), divided by the total number of tidal cycles within the measured period (*cycles total*). This produced a metric similar to the frequency of tidal inundations seen in *Balke et al. (2016)*²⁶. However, in contrast to the inundation frequency metric used there, we then inverted the quantity to measure the percent of *skipped* tidal inundations (Equation 2), which correlates positively with the probability of vegetation establishment (see *Supplementary Fig. 2* for visualization).

$$\textit{Tidal inundations skipped} (\%) = \left(1 - \frac{\textit{cycles inundated}}{\textit{cycles total}}\right) * 100 \% \quad \text{Equation 1}$$

2.8

L178: a nice explanation of the dependence on slope. I thought that "droplets" may be a

poorly chosen word as a model like this does not model droplets. Droplets are controlled largely by surface tension. I think a better word would be water parcels.

R2.8

We agree that this is an important distinction and have changed to term water 'droplet' to water 'parcel' wherever it has appeared in the text.

2.9

L206 what causes the flattening?

R2.9

This comment is in reference to the passage:

'Whereas the transition from bare to vegetated may be regarded as relatively rapid on a geomorphological timescale in line with the alternative stable state framework, at a smaller timescale we can detect a sequence of stages driving the transition. ***This begins with i) the flattening of tidal flats*** which drives ii) the stabilization and intensification of micro-topographic patterning, which then iii) facilitates vegetation establishment.'

The conventional form of an accreting tidal flat develops an increasing convex shape over time. This convex form is made up of a flattening platform in the landward section and a fairly steep transition zone to the lower intertidal and subtidal bathymetry near the channel. We can see evidence of the development of convex tidal flat morphologies throughout the three estuaries in panel Figure 2c, which shows that higher tidal flats are on average flatter.

The standard explanation for the development of a convex tidal flat morphology is that more sediment tends to accrete at the seaward edge of the upper intertidal plate as the flow tide water quickly slows down as it crosses the tidal flat (Friedrichs 2011). As the seaward sections of the tidal flat rise faster than the initially higher landward sections, the upper section of the plate gradually flattens.

We have added an explanation of this process and links to relevant literature to the discussion, replacing the earlier explanation, in the following new passage:

L258:

The pattern of accretion on upper intertidal flats, which is ubiquitous across these three estuaries, is the fundamental driver of marsh expansion. It is most likely that this is a consequence of dredging practices, which enhance the tidal amplitude. When an estuary is dredged, the deeper channel bottom imposes less friction on tidal water entering and exiting the estuary. This enhances the flow velocity in the channel, which in turn leads to tidal amplification as the estuary width narrows along its course upstream (~~van Maren et al. 2015, Stark et al. 2017, van Dijk et al. 2021~~)^{8,12,13}. Fast moving sediment-laden water quickly slows down when it moves onto the comparatively shallow tidal flat during high tide. As the water decelerates, a greater quantity of the sediment is deposited on the tidal flat nearest the channel (~~Femmerman et al. 2004~~)⁴¹. This causes a greater increase in elevation near the channel than near land,

which produces an increasingly convex tidal flat profile over time, with a flat anterior section⁴² (see Supplementary Fig. 4 for an example, de Vet *et al.* 2020). The result of the estuary-scale analysis in this study shows a clear correlation between the slope of the tidal flat and the intensity of micro-topographic patterning (Figure 4a). ...

~~Our analysis demonstrates that similar patterns of heightening and flattening of bare tidal flats preceded transitions to vegetated marshes in three different European estuaries. Currently, the best evidence suggests that this is because the tidal amplitude in these three estuaries has increased in recent decades due to anthropogenic deepening of the estuarine channels for ship navigation and narrowing of the intertidal areas through land reclamation (van Maren *et al.* 2015, Stark *et al.* 2017, van Dijk *et al.* 2021). This increase in tidal amplitude has contributed to gradual build-up via accretion of the remaining intertidal flats above neap high water level (Temmerman *et al.* 2004), which develop into convex morphologies with wide, flat anterior regions (de Vet *et al.* 2020). In this study, we show that atop the high, flattening tidal flats, micro-topographic patterns are more likely to form, which then facilitate the establishment of vegetation. These paired morphological and ecological changes are observed in all three estuaries, which may suggest that these trends represent a general phenomenon in response to anthropogenic deepening and narrowing of estuaries.~~

In addition to this, we have also added the new Supplementary Fig. 4, which gives an archetypal example of an accreting tidal flat that is becoming more convex over time, flattens, develops micro-topography, and eventually becomes vegetated.

Supplementary Figure 4. A showcase of a typical accreting upper intertidal flat that experiences pattern formation and subsequent vegetation establishment. Here we see the development of the tidal flats at Zuidgors and Baarland in the Western Scheldt between 2004 and 2022. In the topmost panels we can see the change in (A) elevation and (B) micro-topographic pattern intensity (SD of NIR-band) between 2004 and 2020. The bottomright panel (C) displays the net change in vegetation cover over the same period. Development of the tidal flat profile can be seen in the panels D & E which correspond to the two transects, depicted by black lines running across the tidal flat (D and E). Put in sequence, panels A - C support our explanation of marsh expansion, where (i) tidal flats rise and flatten as a consequence of higher rates of accretion at the seaward end of the tidal flat, (ii) micro-topographic patterns develop on the raised flattened anterior region, (iii) vegetation is able to expand to a greater extent over areas harboring micro-topographic patterns.

2.10

P269 It is odd to select this as examples .. there are many human activities that affect morphology, like causeways, roading, seawalls, reduced (or increased) sediment supply ...why the focus on dredging

R2.10

This comment refers to the lines:

‘Finding solutions that will mitigate the impact of human activity to the estuary remains extremely difficult. There are two major current propositions to reduce the impact of dredging on estuarine processes: (1) moving inland ports to the coast so that dredging navigation channels is no longer necessary and (2) de-embanking sections of estuaries in ‘managed-realignment’ projects to increase the effective width of the estuary and reduce the tidal amplitude (Turner *et al.* 2007, Stark *et al.* 2017).’

The authors feel that the focus on dredging and embankment is crucial to this story because the effect of this management on the tidal range is the main driver of the geomorphological and ecological changes we describe in this manuscript. Solutions that can modify the tidal range and offset these effects will be important management tools to offset these consequences. We understand that the structure of our original discussion did not lead the reader to this point clearly enough and that our sudden emphasis on dredging seems arbitrary.

To resolve this, we have made significant changes to the structure of these two discussion sections to more clearly develop the role of dredging in our manuscript, as follows (note the first paragraph below is the same as the one that appears in R2.9):

L258:

The pattern of accretion on upper intertidal flats, which is ubiquitous across these three estuaries, is the fundamental driver of marsh expansion. It is most likely that this is a consequence of dredging practices, which enhance the tidal amplitude. When an estuary is dredged, the deeper channel bottom imposes less friction on tidal water entering and exiting the estuary. This enhances the flow velocity in the channel, which in turn leads to tidal amplification as the estuary width narrows along its course upstream (van Maren *et al.* 2015, Stark *et al.* 2017, van Dijk *et al.* 2021)^{8,12,13}. Fast moving sediment-laden water quickly slows down when it moves onto the comparatively shallow tidal flat during high tide. As the water decelerates, a greater quantity of the sediment is deposited on the tidal flat nearest the channel (Temmerman *et al.* 2004)⁴¹. This causes a greater increase in elevation near the channel than near land, which produces an increasingly convex tidal flat profile over time, with a flat anterior section⁴² (see **Supplementary Fig. 4** for an example, de Vet *et al.* 2020). The result of the estuary-scale analysis in this study shows a clear correlation between the slope of the tidal flat and the intensity of micro-topographic patterning (**Figure 4a**). The results of our ‘inclined plane’ model suggest that the formation of micro-topographic patterns may be related to the formation of pools and the concentrating flow pathways of draining surface water in nearly flat environments (**Figure 4**). While this model provides a robust explanation for why patterns do not form in steeper areas, it cannot provide a complete explanation of the conditions required for patterns to form. It cannot, for instance, explain the decrease in micro-topographic

intensity in near-zero slope environments that is measured in the geospatial data. We have also found that micro-topographic patterning is more intense when the tidal flat is frequently above water during high tide (Figure 4b). This suggests that the low tide period itself may play an important role in the amplification of these patterns, perhaps by exposing the sediment to prolonged periods of drying. It is also worth considering that the widening and heightening upper intertidal flat will also experience a reduced wave climate, which will reduce sediment turn-over (Friedrichs 2011)¹⁴. Each of these various environmental characteristics in combination are likely important to the formation of permanent micro-topographic patterns. The major weakness in our understanding of these patterns up to this point is that has so far been guided exclusively by correlative evidence.

L325:

What is to come of these changes? ~~Once estuarine morphology begins to facilitate the expansion of vegetation, consequences for both humans and the ecology will follow.~~ On its face, a net gain in vegetated foreshore area would appear to benefit flood safety (Temmerman *et al.* 2013, Bouma *et al.* 2014, Zhu *et al.* 2020)^{17,18,48}. However, in already narrow estuaries there is some risk that if marshes advance seaward they would further raise the height of colonized flats by further enhancing sediment accretion (Wang *et al.* 2020)³¹. The creation of high intertidal areas that are less often submerged effectively decreases the width of the estuary during a large period of the tidal cycle. This could cause the propagation and magnification of the tidal wave upstream (Broekx *et al.* 2011, Stark *et al.* 2017, Leuven *et al.* 2019)^{2,13}, putting upstream communities at greater flood risk. This is somewhat ironic, given that wider marshes enhance flood security locally by reducing wave impact and run-up on the dikes behind them (Zhu *et al.* 2020)⁴⁸. Greater marsh area will also come at the cost of feeding habitat for migratory birds in the estuary due to narrowing of the intermediate intertidal zones between the central channel and the marsh (Strahlberg *et al.* 2010, Yang *et al.* 2011, Strong & Ayres 2016, Mu & Wilcove 2020)^{19,49,50}. Altogether, an intertidal system composed pre-dominantly of high intertidal flats bisected by a single deep central channel would host a poorer diversity of intertidal habitats and biodiversity (Reise 2005, Cozzoli *et al.* 2017)^{9,51}. Lastly, there may also be long-term risks associated with the development of the estuary in this direction. Studies in micro- and meso-tidal habitats have shown that large vegetated marsh interiors are known to be more prone to drowning (Kirwan *et al.* 2010, Schepers *et al.* 2017, Silvestri *et al.* 2018)^{52,53,54}, which may become a risk in macro-tidal estuaries once sea-level rise begins to accelerate in this region (Leuven *et al.* 2019)².

2.11

I am not so keen on the last 2 paragraphs of the discussion. They do not really add a lot of insight. It would be better to provide recommendations on data needed to explore their hypothesis more thoroughly.

R 2.11

We're sorry to hear that and hope now that with the changes in structure of our discussion described in R2.10 you will find our last discussion paragraphs a better fit for the article in general.

Reviewer #3 (Remarks to the Author):

This manuscript presents a novel way of forecasting the establishment of vegetation in tidal flats, and thus its transition to a marsh, using the presence and intensity of micro-topography within the tidal flat. By linking the slope of the tidal flat to the development of micro-topography, the authors propose simple geomorphic metrics to describe the state change. This is supported by a neat process-based conceptual picture.

The topic is certainly interesting and relevant, and the scale and depth of the data analyzed fits the scope of the conclusions. However, there are several inconsistencies in the way the data is presented and shown in the figures that undermine authors' conclusions. Therefore, I cannot recommend publication until those are properly resolved or clarified.

Major points:

3.1

1. A key conclusion is that the intensity (or degree of development) of micro-topography is crucial for the development of relatively low-elevation marshes. As shown in Fig.3, this is supported by the dependency of the probability of new vegetation on the two explanatory variables: (1) the intensity of micro-topography and (2) the percentage of tidal inundations skipped. However, as the authors state (and show in Fig.S3), they are not independent variables and a more intense micro-topography correlates with higher elevation. Since it is well known that higher elevation leads to new vegetation, that would contradict the relevance of Fig.3 and authors' conclusion.

R3.1

This is an important concern. In order to address it, we have performed a more in-depth analysis of the distribution of tidal flats, displayed in the new Supplementary Fig. 3 (see below). In this figure, we show how the average micro-topographic intensity changes across tidal flats of different slope and elevation. In Supplementary Fig. 3c, we display the correlation between elevation and micro-topography and see that, as Review 3 noted, micro-topography formation is more likely to occur at higher elevation. However, if we also consider the slope of the tidal flat, as in the new panel Supplementary Fig. 3a, we can see that high elevation alone does not generally lead to pattern formation. Instead, a low tidal flat slope is required for pattern formation to occur across the entire elevation range. This can be seen by the vertical, fairly rectangular shape of the darker region in Supplementary Fig. 3a, which shows that micro-topography becomes much stronger on tidal flats shallower than ~ 0.2 degrees (also see figure annotations). This suggests that *both* sufficient elevation (areas above neap low water) and sufficiently flat slope (below ~ 0.2 degrees) are dual pre-requisites of pattern formation.

Furthermore, Figure 3 also provides strong evidence that there is a complementary facilitative effect of both micro-topographic intensity and elevation on marsh expansion in these estuaries. If micro-topography played no role in marsh establishment (and elevation was the only important factor), we would expect to see a horizontal boundary between the light yellow unvegetated area and the dark vegetated area in Figure 3. Instead, we see a clear diagonal boundary, indicating that with stronger micro-topographic patterning, vegetation can establish lower in the intertidal. We have now updated the color scheme of Figure 3 to accentuate these differences. In the newly added Supplementary Fig. 9, which is an elaborated version of the main manuscript Figure 3, we also display the separate positive effects of elevation and micro-topography on marsh establishment (panels C & D).

In addition to these figures, we have altered the results & discussion to further explore this topic in the following passages:

Results

L174:

Micro-topography forms on gently sloping upper intertidal flats

By using lidar data and false color images in combination to detect very small-scale topographic features, we determined that micro-topographic patterns are found predominantly on tidal flats with very shallow slopes, peaking in intensity at around 0.1 degrees (Supplementary Fig. 3d). As with vegetation establishment, the intensity of tidal flat patterning is also consistently stronger in areas that experience more frequent relief from tidal inundation. (Supplementary Fig. 3c). However, no matter the elevation, micro-topographic patterns tend not to occur on tidal flats with a slope greater than 0.3 degrees (see Supplementary Fig. 3a). ~~Micro-topographic patterns occur much more frequently on tidal flats with a slope less than ~0.5 degrees. Like vegetation, these patterns also seem to occur preferentially above neap high water and decrease sharply in frequency in areas that experience less frequent relief from tidal inundation (Figure S3).~~ Thus, patterns can be expected to appear most often on accreting, convex upper intertidal flats, which feature flat anterior sections high in the tidal frame (see Supplementary Fig. 4 for an example).

Discussion

L269:

The result of the estuary-scale analysis in this study shows a clear correlation between the slope of the tidal flat and the intensity of micro-topographic patterning (Figure 4a). [...] We have also found that micro-topographic patterning is more intense when the tidal flat is frequently above water during high tide (Figure 4b). This suggests that the low tide period itself may play an important role in the amplification of these patterns, perhaps by exposing the sediment to prolonged periods of drying. It is also worth considering that the widening and heightening upper intertidal flat will also experience a reduced wave climate, which will reduce sediment turn-over (Friedrichs 2011)¹⁴. Each of these various environmental characteristics in combination are likely important to the formation of permanent micro-topographic patterns. The major weakness in our understanding of these patterns up to this point is that has so far been guided exclusively by correlative evidence.

We still cannot yet provide a complete explanation of how and why these high, low-slope tidal flat environments tend to produce micro-topographic bed level variation, but we can provide some direction for further inquiry: [...]

New figures (for quick reference for the editor & reviewers, note these figures were also introduced earlier in R1.5 & R2.1)

Supplementary Figure 3. The effects of tidal flat slope and position in the intertidal on micro-topographic intensity are displayed in four panels, considering all three studied estuaries. In Panel A, the interactive effects of tidal position and slope on micro-topographic intensity are explored over a 2D field. Here darker colors represent on average more intense micro-topographic patterning. Panel B shows the number of observations at each cell in the field (cells supported by less than 30 observations have been removed). Note the strong correlation between high position in the intertidal and low slope tidal flats, indicated by the distribution of observations across this frame. Panels C and D display the effects of tidal inundation (C) and slope (D) independently. Here, points represent the average of a binned group and error bars show standard error. These panels show that micro-topography is stronger when higher in the tidal frame on very shallow slopes. However, panel A shows that micro-topography only occurs in the upper intertidal within low slope regions (slopes less than 0.2 - 0.3 degrees), indicating that slope may be the predominant precursor of pattern formation.

Supplementary Figure 9. An extended version of Figure 3. In addition to the original figure (A), here we also display the total number of observations represented within each cell in the raster (B) and separate the effects of the tidal position (panel C) and the intensity of micro-topography (D) on vegetation establishment into 1D scatterplot figures. The points in panels C and D represent the mean value within each binned group, and the error bars show standard error.

3.2

2. The range of the micro-topography metric (SD of NIR band) is different in all the figures: in Fig.S3 it goes from ~4 to ~7; in Fig.4A, from ~3.5 to ~4.5; and in Fig.3, from 1 to 50. This is relevant because, as shown in Fig.3, the range above 7 is the one better predicting vegetation establishment at lower elevations, but this range is excluded from the comparison with slope (Fig.4) and tidal inundations skipped (Fig.S3). Without this information, there is no way to verify the actual relation between tidal flat slope and marsh establishment, or properly check the range where the explanatory variables are actually independent (see point above).

R3.2

The authors believe this conclusion is the result of an understandable misinterpretation of the original versions of Figures 4 and S3. We hope that changes to these figures detailed here will resolve this issue.

In Figure 3, we see that measured values of micro-topographic intensity can have a broad range between 1 and nearly 50. But the points in the original version's Figures 4 and S3 show the average value of a binned group and thus have a much smaller range of values, between approximately 3 and 11. It makes sense that this caused confusion since we originally did not include error bars to communicate that these points represented mean values.

In the updated version of Figure S3 (Supplementary Fig. 3), we have combined panel A of the original Figure 4 with Figure S3 (micro-topography vs tidal inundations skipped). Both of these panels now display points with standard error bars and the y-axes now display the two figures over the same range of values to make comparisons between their effects clearer. Note that these points now display the mean values while in the earlier version they had represented the median. Note also that wherever we refer to the average intensity of a binned group in a figure, the axis label has been adjusted to read 'Avg. intensity of micro-topography' (as in Fig. 4 & Supplementary Fig. 3), rather than 'Intensity of micro-topography' when it refers to individual measured values (as in Fig. 3).

As mentioned above (in R3.1), we have chosen to further explore the causes of pattern formation by combining these factors first explored in two 1D figures (panels C & D) into a single 2D raster to visualize the interactive role of elevation and slope on pattern formation (Figure 4a). This is equivalent to how we chose to explore vegetation expansion in Figure 3.

We hope that with this improved data visualization we will convince Reviewer 3 especially that although pattern formation is clearly linked to high position in the intertidal, the slope of the mudflat cannot be ignored as a key environmental characteristic allowing for pattern formation. In particular, notice the annotated region on Figure 4a (see above in R3.1), which

highlights the lack of micro-topography throughout the upper intertidal in areas that are too 'steep' (> 0.2-0.3 degrees).

3.3

3. The most important explanatory variable, the micro-topography metric (defined as the standard deviation of the NIR band), is never calibrated. There is no way to know what is the lower bound for the absence of micro-topography or how it increases with the actual micro-topography relief. For example, in Fig.4A the authors imply that a micro-topography metric less than 3.5 represents the absence of micro-topography (for large enough slopes). However, this is not obvious from Fig.3, where there are points with a micro-topography metric as low as 1.

R3.3

It's obvious this validation needs to be performed and we apologize that this was not included in the original manuscript. Please see our resolution to this problem detailed in our response to reviewer 1 on this topic: R1.1.

Minor suggestions:

3.4

4. Can you add another plot with the frequency of each pair (x,y) in Fig.3 to better illustrate the relation between the two explanatory variables?

R3.4

Yes, you can find this requested figure in panel B of the new Supplementary Figure 9 (see R2.1).

3.5

5. In Fig. 4A, either add the standard deviation of the micro-topography metric for each slope bin or create a scatter plot (or 2D histogram) with all data, to better understand the relation with local slope.

R3.5

This change has been made. Please see our full response to this request in R3.2.

3.6

6. In Fig.S2, add a plot as panel C but vs. micro-topography metric (perhaps for tidal inundations skipped < 60%) to better understand the average relation between new vegetation and micro-topography without considering local elevation.

R3.6

In Supplementary Figure 2c, we display the probability that vegetation is *present* against the position in the tidal frame (tidal inundations skipped %). Unfortunately, it is not possible to compare mudflat micro-topography with where vegetation *currently is*, because the presence of vegetation changes the values of the micro-topography. When measured with photos, the vegetation simply increases the standard deviation in the image, giving a falsely high reading. Even when using lidar of sufficiently fine resolution (such as with our terrestrial lidar campaigns, see Supplementary Figure 7) and measuring slopes, the presence of vegetation can still cause artifacts that lead to enhanced standard deviation, and enacting real changes to the tidal flat via biogeomorphic feedbacks so that it does not continue to represent conditions when the vegetation first appeared.

Instead, throughout this manuscript we have compared the micro-topography of the tidal flat in years *directly preceding* the expansion of vegetation (probability of vegetation *expansion*, rather than *presence*).

In order to meet the request of Reviewer 3, we added a figure similar to the one requested, but using vegetation expansion rather than presence. This can be found in panel D of Supplementary Figure 9 (see R2.1).

3.7

7. Line 314 - It is not clear what do you mean by "inverted".

R3.7 We have tried to clarify the meaning of 'inverted' here by adding the new Equation 1, which provides a visual representation of our description in the preceding passage. For clarity 'inverted' refers to the '1 - ' term in the equation, which takes the

'inundations experienced' term: $\left(\frac{\text{cycles inundated}}{\text{cycles total}} \right)$

and transforms it into an 'inundations not experienced' term:

$$\left(1 - \frac{\text{cycles inundated}}{\text{cycles total}} \right)$$

See the text changes below:

'These water level data were used to calculate the frequency of tidal inundations at every intertidal elevation by calculating the number of high tides that reached above a given elevation (*cycles inundated*), divided by the total number of tidal cycles within the measured period (*cycles total*). This produced a metric similar to the frequency of tidal inundations seen in Balke *et al.* (2016)²⁶. However, in contrast to the

inundation frequency metric used there, we then inverted the quantity to measure the percent of *skipped* tidal inundations (Equation 2), which correlates positively with the probability of vegetation establishment (see Supplementary Fig. 2 for visualization).

$$\text{Tidal inundations skipped (\%)} = \left(1 - \frac{\text{cycles}_{\text{inundated}}}{\text{cycles}_{\text{total}}}\right) * 100 \% \quad \text{Equation 1'}$$

Lastly, we provide a list of further changes made to meet **Nature Communications'** **formatting guidelines:**

Abstract shortened from 178 words to 150 words

'Abstract

Forecasting transitions between tidal ecosystem states, such as between bare tidal flats and vegetated marshes, is crucial because may imply irreversible loss of highly valued ecosystem services. In this study, we combine geospatial analyses of three European estuaries with a simple numerical model to demonstrate that the development of 'micro-topographic' patterning on tidal flats is a robust early indicator of marsh establishment. We first show that the development of micro-topographic patterns precedes vegetation establishment, and that these patterns tend to form only on tidal flats with a slope of $<0.5 < 0.3$ degrees. ~~Second, numerical modelling suggests that micro-topography develops due to the concentration of draining surface water into united flow pathways, which is maximized on very gentle slopes.~~ Second, numerical modelling provides an explanation for the formation of micro-topography due to the natural concentration of draining surface water on very gentle slopes. ~~As a consequence, the slope of a tidal flat and subsequent micro-topographic development can be considered early warning signals of transitions from bare tidal flat to vegetated marsh.~~ We find this early indicator to be robust across three estuaries where anthropogenic deepening and narrowing has occurred in recent decades, which may suggest its broader applicability to other estuaries with similar morphological management.'

Subheadings in the Results & Methods shortened to max 60 characters:

Results:

L148:

~~*Tidal flats are flattening as they reach higher elevations across European estuaries*~~

Tidal flats flatten as they become higher across European estuaries

L158:

~~*Pioneer vegetation is more likely to establish at lower elevation when micro-topographic patterns are present*~~

Pioneer vegetation establishment on micro-topographic patterns

L174:

~~*Micro-topographic patterns intensify on very shallowly sloping upper intertidal flats*~~

Micro-topography forms on gently sloping upper intertidal flats

L207:

~~Modelling indicates that micro-topography forms as a consequence of the concentration of surface water~~

Micro-topography may arise from patterns in draining surface water

Methods:

L393:

~~Frequency of tidal immersion from bathymetry data & water level time series~~

Calculating the frequency of skipped tidal inundations

L480:

~~Distinguishing micro-topographic patterns from vegetation and other bathymetric patterns~~

Distinguishing micro-topographic patterns from other features

L500:

~~Long-term changes in intertidal elevation, slope, and vegetation cover~~

Changes in intertidal elevation, slope, and vegetation cover

L523:

~~Facilitation of vegetation establishment by tidal elevation & micro-topography~~

Effect of elevation & micro-topography on vegetation establishment

L544:

~~Relationship between underlying slope and the occurrence of micro-topography~~

Role of elevation & slope in formation of micro-topography

Added data & code availability sections:

Data Availability

The geospatial data and scripts used to perform the analyses that support the findings of this study are archived and publicly available via 4TU.Research Data, doi:10.4121/21762680. Source data for figures 2 - 4, and supplementary figures 1-5, 7, and 9 are provided with the paper.

Code Availability

The code used to run the ‘inclined plane’ numerical model that supports the findings of this study is archived and publicly available via 4TU.Research Data, doi:10.4121/21762680.

References:

- Removed 20 references to achieve the 70 reference limit. These references were chosen for omission because they were conceivably redundant with other cited articles and appeared in series with other citations.

The omitted citations include:

Best ÜSN, van der Wegen M, Dijkstra J, Willemsen PWJM, Borsje BW, Roelvink DJA. 2018. Do marshes survive sea level rise? Modelling wave action, morphodynamics and vegetation dynamics. *Environmental Modelling & Software* 109: 152–166. doi:10.1016/j.envsoft.2018.08.004.

Borchert, SM, Osland, MJ, Enwright, NM, Griffith, KT. 2018. Coastal wetland adaptation to sea level rise: Quantifying potential for landward migration and coastal squeeze. *J Appl Ecol.* 55: 2876–2887. doi:10.1111/1365-2664.13169.

Callaghan DP, Bouma TJ, Klaassen P, van der Wal D, Stive MJF, Herman PMJ. 2010. Hydrodynamic forcing on salt marsh development: Distinguishing the relative importance of waves and tidal flows. *Estuarine, Coastal and Shelf Science* 89(1): 73–88. doi:10.1016/j.ecss.2010.05.013.

Chaussard E, Amelung F, Abidin H, Hong S-H. 2013. Sinking cities in Indonesia: ALOS PALSAR detects rapid subsidence due to groundwater and gas extraction. *Remote Sensing of Environment* 128: 150–161. doi:10.1016/j.rse.2012.10.015.

Corenblit D, Steiger J, Gurnell AM, Tabacchi E, Roques L. 2009. Control of sediment dynamics by vegetation as a key function driving biogeomorphic succession within fluvial corridors. *Earth Surf. Process. Landforms*, 34: 1790–1810. doi:10.1002/esp.1876.

van der Heide T, van Nes EH, Geerling GW *et al.* 2007. Positive Feedbacks in Seagrass Ecosystems: Implications for Success in Conservation and Restoration. *Ecosystems*, 10:1311. doi:10.1007/s10021-007-9099-7.

Kirwan M, Temmerman S, Skeeahan E, *et al.* 2016. Overestimation of marsh vulnerability to sea level rise. *Nature Clim Change* 6, 253–260. doi:10.1038/nclimate2909.

Kondolf GM, Rubin ZK, Minear JT. 2014. Dams on the Mekong: Cumulative sediment starvation. *Water Resour. Res.* 50: 5158–5169. doi:10.1002/2013WR014651.

van de Koppel J, van der Wal D, Bakker J, Herman PMJ. 2005. Self-Organisation and Vegetation Collapse in Marsh Ecosystems. *The American naturalist*. 165: E1–12. doi:10.1086/426602.

Liu, Z., Fagherazzi, S. & Cui, B. 2021. Success of coastal wetlands restoration is driven by sediment availability. *Commun Earth Environ* 2: 44. doi:10.1038/s43247-021-00117-7.

Maina J, de Moel H, Zinke J, *et al.* 2013. Human deforestation outweighs future climate change impacts of sedimentation on coral reefs. *Nat Commun* 4: 1986. doi:10.1038/ncomms2986.

Maina Gichaba C, Kipseba EK, Masibo M. 2013. Chapter 20—Overview of Landslide Occurrences in Kenya: Causes, Mitigation, and Challenges. Paron P, Olago DO, Omuto CT (Ed.). *Developments in Earth Surface Processes*. Elsevier, 16: 293–314. doi:10.1016/B978-0-444-59559-1.00020-7.

Moffett KB, Nardin W, Silvestri S, Wang C, Temmerman S. 2015. Multiple stable states and catastrophic shifts in coastal wetlands: Progress, challenges, and opportunities in validating theory using remote sensing and other methods. *Remote Sensing*, 7: 10184–10226. doi:10.3390/rs70810184.

Mohamadi MA, Kaviani A. 2015. Effects of rainfall patterns on runoff and soil erosion in field plots. *International Soil and Water Conservation Research* 3(4): 273–281. doi:10.1016/j.iswcr.2015.10.001.

Qian Zhang, Zheng Gong, Changkuan Zhang, Ian Townend, Chuang Jin, Huan Li. 2016. Velocity and sediment surge: What do we see at times of very shallow water on intertidal tidal flats? *Continental Shelf Research*. 113: 10–20. doi:10.1016/j.csr.2015.12.003.

Salman A, Lombardo S, Doody JP. 2004. Living with coastal erosion in Europe: Sediment and Space for Sustainability. PART 1 Major finding and Policy recommendations of the EUROSION project 10 May 2004. Service contract B4 3301/2001/329175/MAR/B3 “Coastal erosion Evaluation of the need for action” Directorate General Environment. European Commission, Brussels, BE.

Silinski A, Heuner M, Troch P, *et al.* 2016. Effects of contrasting wave conditions on scour and drag on pioneer marsh plants. *Geomorphology* 255: 49–62. doi:10.1016/j.geomorph.2015.11.021.

Wang S, Fu B, Gao G, Liu Y, Zhou J. 2013. Responses of soil moisture in different land cover types to rainfall events in a re-vegetation catchment area of the Loess Plateau, China. *CATENA* 101: 122–128. doi:10.1016/j.catena.2012.10.006.

Xie T, Cui B, Li S, Bai J. 2019. Topography regulates edaphic suitability for seedling establishment associated with tidal elevation in coastal marshes. *Geoderma* 337: 1258–1266. doi:10.1016/j.geoderma.2018.07.053.

Yang SL, Zhang J, Zhu J, Smith JP, Dai SB, Gao A, Li P. 2005. Impact of dams on Yangtze River sediment supply to the sea and delta intertidal wetland response. *J. Geophys. Res.* 110: F03006. doi:10.1029/2004JF000271.

We also,

- Re-ordered references in order of appearance in the text.
- Replaced (Author et al., year) ref. style with the Nature journal’s numbered ref. style.

Added Author contributions & competing interests sections:

Author Contributions

G.S.F and T.J.B. conceived the ideas. G.S.F designed the methodology, performed the analyses of results, and wrote and ran the simulations. G.S.F. and M. H. performed data acquisition. G.S.F., S.T., M.K. and T.J.B. led the writing of the manuscript. All authors contributed critically to the drafts and gave final approval for publication.

Competing Interests

The authors declare no competing interests.

Figures:

The structure of the figure captions has been updated to align with the *Nat Comm* formatting guidelines. The first sentence of all figures now appears in bold and provides an overview of the figure. See below:

(The order numbering of the figures has also been updated to account for the new figures.)

*Figure 1. **A visual showcase of the role of micro-topographic patterns in vegetation establishment.***

*Figure 2. **Changes in upper intertidal flat morphology across three study estuaries.***

*Figure 3. **Correlation between elevation, micro-topography, and vegetation establishment.***

*Figure 4. **Comparison of model results with the distribution of micro-topographic patterns in the geospatial data.***

*Supplementary Figure 1. **The change in the vegetated area of the Humber, Western Scheldt, and Elbe over the years of data availability fit to linear regressions.***

*Supplementary Figure 2. **A schematic illustration explaining the origin and calculation of the ‘tidal inundations skipped (%)’ metric.***

*Supplementary Figure 3. **The effects of tidal flat slope and position in the intertidal on micro-topographic intensity are displayed in four panels, considering all three studied estuaries.***

*Supplementary Figure 4. **A showcase of a typical accreting upper intertidal flat that experiences pattern formation and subsequent vegetation establishment.***

*Supplementary Figure 5. **The frequency of different NDVI values over the upper intertidal (above neap high water level) are displayed in four panels.***

*Supplementary Figure 6. **This figure showcases the utility of using orthophotos to quantify micro-topographic patterns, rather than relying on relatively coarser resolution lidar-derived elevation maps.***

*Supplementary Figure 7. **A calibration of the technique of using photos to measure micro-topography, compared with elevation measurements using terrestrial lidar.***

*Supplementary Figure 8. **This figure depicts the various regions of the estuary that are masked (removed) before the quantification of micro-topography.***

*Supplementary Figure 9. **An extended version of Figure 3.***

The names of supplementary appendices 1 – 4 have been changed to:

- (1) **Supplementary Methods - Data acquisition and pre-processing**
- (2) **Supplementary Table 1**
- (3) **Supplementary Table 2**
- (4) **Supplementary Note - Public data portals**

REVIEWERS' COMMENTS

Reviewer #1 (Remarks to the Author):

I believe the authors did a very good job addressing all of my concerns of the original manuscript (as well as the 2 other reviewers) within this revised manuscript. They obviously spent considerable time reanalyzing data, adding new figures, and revising text, and I believe they have created a greatly improved manuscript that will be of high interest to coastal geomorphologists and ecologists. I have no additional comments or concerns.

Reviewer #2 (Remarks to the Author):

The changes that have been made to the manuscript are comprehensive. I can clearly see the link between vegetation and microtopography, and see that this link exists even if the elevation is held constant. The modelling is improved, although still is very much in an exploratory stage. The authors have shown that the modelling results are dependent on the way in which the model is set up. (More sophisticated modelling will not change that outcome, so as long as the sensitivity is noted, it is convincing enough to proceed). It is clear that the outcome is relevant for this particular site (with no seagrass/mangroves etc) and further work will be needed to determine whether the finding is relevant for more diverse sites. The authors have made this clear in the revised manuscript. I have no further issues with the manuscript.

Reviewer #3 (Remarks to the Author):

The authors successfully addressed all my concerns and comments. Therefore, I'm happy to recommend the publication of this manuscript.

On behalf of all the authors, I would like to thank all three reviewers and the editor for their excellent feedback on our manuscript. We too think that the manuscript has improved drastically through the revision process, and I am grateful to have had such interested and critical reviewers.

REVIEWERS' COMMENTS

Reviewer #1 (Remarks to the Author):

I believe the authors did a very good job addressing all of my concerns of the original manuscript (as well as the 2 other reviewers) within this revised manuscript. They obviously spent considerable time reanalyzing data, adding new figures, and revising text, and I believe they have created a greatly improved manuscript that will be of high interest to coastal geomorphologists and ecologists. I have no additional comments or concerns.

Reviewer #2 (Remarks to the Author):

The changes that have been made to the manuscript are comprehensive. I can clearly see the link between vegetation and microtopography, and see that this link exists even if the elevation is held constant. The modelling is improved, although still is very much in an exploratory stage. The authors have shown that the modelling results are dependent on the way in which the model is set up. (More sophisticated modelling will not change that outcome, so as long as the sensitivity is noted, it is convincing enough to proceed). It is clear that the outcome is relevant for this particular site (with no seagrass/mangroves etc) and further work will be needed to determine whether the finding is relevant for more diverse sites. The authors have made this clear in the revised manuscript. I have no further issues with the manuscript.

Reviewer #3 (Remarks to the Author):

The authors successfully addressed all my concerns and comments. Therefore, I'm happy to recommend the publication of this manuscript.